# SparseVLM: Visual Token Sparsification for Efficient Vision-Language Model Inference

## Abstract

In vision-language models (VLMs), visual tokens usually consume a significant amount of computational overhead, despite their sparser information density compared to text tokens. To address this, most existing methods learn a network to prune redundant visual tokens and require additional training data. Differently, we propose an efficient training-free token optimization mechanism dubbed SparseVLM without extra parameters or fine-tuning costs. Concretely, given that visual tokens complement text tokens in VLMs for linguistic reasoning, we select visual-relevant text tokens to rate the significance of vision tokens within the self-attention matrix extracted from the VLMs. Then we progressively prune irrelevant tokens. To maximize sparsity while retaining essential information, we introduce a rank-based strategy to adaptively determine the sparsification ratio for each layer, alongside a token recycling method that compresses pruned tokens into more compact representations. Experimental results show that our SparseVLM improves the efficiency of various VLMs across a range of image and video understanding tasks. In particular, LLaVA equipped with SparseVLM reduces $61\% \sim 67\%$ FLOPs with a compression ratio of 78% while maintaining 93% of the accuracy.

## 1 Introduction

Benefiting from tremendous advancements in large language models (LLMs) (Radford et al., 2019; Brown et al., 2020; Achiam et al., 2023; Touvron et al., 2023; Peng et al., 2023; Bi et al., 2024), the realm of vision-language models (VLMs) has undergone a revolutionary progress. To combine visual signals with textual semantics, the mainstream practice in VLMs (Team et al., 2023; Bai et al., 2023b; Chen et al., 2023; Li et al., 2024b; 2023a) employs sequential visual representation, where images are extracted into vision tokens and sent into an LLM decoder. With modal alignment and instruction fine-tuning (Du et al., 2021; Liu et al., 2023a; Zhu et al., 2023b), recent VLMs successfully adapt LLMs to the vision domain and inherit their perception and reasoning abilities.

Despite the promising performance, further incorporation of visual tokens inevitably introduces a huge memory and computation overhead when compared to LLMs, particularly for high-resolution images (Li et al., 2024b) and multi-frame videos (Lin et al., 2023). For instance, a $672 \times 672$ image in LLaVA (Liu et al., 2024) yields 2304 vision tokens that span over half of the context length. However, the information in images is typically more sparse than in natural languages (Marr, 2010), resulting in inefficiency in directly processing both modalities. To address this, existing methods extract more compact image representations by modifying the image encoder or projector (Alayrac et al., 2022; Li et al., 2024a; Dai et al., 2023; Cha et al., 2024). While some recent works further sparsify vision tokens during the decoding (Ye et al., 2024; Chen et al., 2024b; Cai et al., 2024), they still ignore the guidance from the language tokens, which contradicts the multimodality paradigm. We argue that *visual tokens should be sparsified adaptively based on the question prompt*, as the model might focus on different parts (e.g., foreground or background) when dealing with various questions, as shown in Figure 1. Furthermore, current approaches generally train a network to prune redundant visual tokens and require additional training data (Li et al., 2024a; Ye et al., 2024).

In this paper, we introduce a **text-guided training-free** framework dubbed **SparseVLM** for efficient vision language model inference. We reuse the self-attention matrix of visual-text tokens directly from the decoder layers without extra training parameters for sparsification. We ascertain that *not all prompt tokens should be considered* as some could be less relevant, which leads to inaccurate

Figure 1: **Visualization of different visual token sparsification methods.** Unlike previous methods with text-agnostic visual sparsification (c) e.g., recent VocoLLaMA (Ye et al., 2024), our SparseVLM (b) is guided by question prompts to select relevant visual patches. Best viewed in color.

correlation results and downgrades the performance of sparse inference. Specifically, our Sparse-VLM first identifies text tokens strongly correlated with visual signals via cross-attention. Then, we measure the contribution of visual tokens to the selected visual-relevant text tokens (raters) and adaptively prune the insignificant vision tokens. Instead of directly discarding the pruned tokens, we further recycle and cluster them to reconstruct more compact tokens to minimize information loss. Due to the information density varying for different image inputs, we employ the rank of the attention matrix to indicate the redundancy level and set an adaptive sparsification ratio accordingly.

The proposed method is simple yet practical. It can act as a plug-and-play module to improve the efficiency of VLMs without additional fine-tuning. Extensive experiments demonstrate that our SparseVLM effectively reduces the computation of various VLMs without sacrificing their performance in a wide range of image and video understanding tasks. For example, LLaVA (Liu et al., 2024) equipped with SparseVLM achieves a $4.5\times$ compression rate while maintaining $93\%$ of its original performance. Alternatively, the latency (CUDA time) can decrease by $53.9\%$ with only a $13\%$ drop in accuracy. To investigate the effectiveness of our method in the video tasks, we further apply SparseVLM to VideoLLaVA (Lin et al., 2023) to additionally compress temporal frames. Without complex design changes, SparseVLM can sparsify video frames into an adaptive number of vision tokens and outperforms existing vision compression methods in video question-answering benchmarks. For instance, our method average exceeds FastV (Chen et al., 2024b) by $34.4\%$.

In summary, the contributions of this paper are threefold:

1. We introduce a novel sparsification framework dubbed SparseVLM for vision-language models. To the best of our knowledge, it is the first attempt to explore the potential of text-aware guidance for efficient inference of VLMs, where additional training is unnecessary.

2. In the framework, we first assign visual-relevant text tokens as raters, to judge the importance of vision tokens. Additionally, the rank of the attention logits is employed to reflect the redundancy and adaptively prune VLMs. Finally, we recycle partial tokens from the pruned pools and reconstruct them to accommodate more information within fewer slots.

3. We apply our SparseVLM framework to both image and video VLMs and conduct extensive experiments across various benchmarks. Our approach consistently outperforms the existing state-of-the-art method FastV by $7.7\% \sim 14.8\%$ on LLaVA, $10.2\% \sim 21.6\%$ on MiniGemini, and $34.4\%$ on VideoLLaVA.

## 2   RELATED WORK

**Vision-Language Models.** With the impressive success of large language models (LLMs) (Achiam et al., 2023; Touvron et al., 2023; Bai et al., 2023a), recent works on generative vision-language models (VLMs) (Liu et al., 2023a; Chen et al., 2023; Li et al., 2024b) improve multimodal comprehension and generation by receiving a long visual token sequence. Moreover, processing higher-resolution images inevitably entails an exponential growth in the length of the visual sequence. For

Figure 2: **The architecture of SparseVLM.** In stage (a), text raters are pre-selected before entering the sparsification LLM. In stage (b), adaptive sparsification is performed on LLM layers, involving computing redundancy and the recycling of reconstructed tokens. Best viewed in color.

example, LLaVA encodes $336 \times 336$ images into 576 tokens (Liu et al., 2024) and processes images with a greater resolution of $672 \times 672$ into 2880 tokens (Liu et al., 2023a). Along the same lines, mini-Gemini-HD (Li et al., 2024b) converts images into 2880 vision tokens based on the standard of high resolution $1536 \times 1536$ and low resolution $672 \times 672$. Comprehending videos or multiple images inherently necessitates increased token slots for visual signals. The VideoLLaVA (Lin et al., 2023) and VideoPoet (Kondratyuk et al., 2023) models allocate thousands of tokens to afford the representation of multiple frames. However, the large number of vision tokens leads to a huge bottleneck for computational infrastructure. Further research and development in sparsification technologies are urged to overcome these hurdles and fully unleash the potential of VLMs.

**Visual Compression for VLMs.** Compression of vision tokens is necessary because, on the one hand, their quantity is usually tens to hundreds of times that of language tokens. On the other hand, visual signals are inherently redundant compared to dense human-designed texts (Marr, 2010). Past efforts to address the above problem can be categorized into two directions. The first one centers on the vision tower or projection of visual modality and cuts vision tokens with external modules. For instance, LLaMA-VID (Li et al., 2024a) exploits the use of Q-Former with context token while DeCo (Yao et al., 2024) employs an adaptive pooling to downsample the visual tokens at the patch level. The other type methods (Ye et al., 2024; Chen et al., 2024b; Cai et al., 2024) go deeper into the text modality and sparsify vision tokens during LLM decoding, but they still neglect the guidance from the text tokens. In our paper, SparseVLM takes note of this and improves performance upon it.

**Token Merging for VLMs.** Token merging has recently received much attention in the field of VLMs where its algorithms mainly fall into two directions. One focuses on the matching algorithm using the Bipartite Soft Matching (BSM) (Bolya et al., 2022). For example, ToMe (Bolya et al., 2022) merges similar visual patches in Transformer blocks and speeds up the match process through the Bipartite Soft Matching (BSM) algorithm. Other methods rely on clustering methods for token merging. For instance, LLaVolta (Chen et al., 2024a) proposes the visual context compressor (an average pooling) to merge the output tokens from the vision tower and progressively enhance the VLMs' efficiency by training. Inspired by TCFormer (Zeng et al., 2024), we apply $k$-nearest neighbor density peak aggregation algorithm (Rodriguez, 2014) to the field of vision-language models. Unlike TCFormer, our method focuses on merging the dropped tokens and makes full use of their detailed information.

## 3 PROPOSED APPROACH: SPARSEVLM

In this section, we present our SparseVLM framework for efficient VLM inference. We first review the attention mechanism in VLMs and then introduce the detailed strategies for our visual sparsification, including visual significance estimation, relevant text token selection, and sparsification level adaptation. We further propose the token recycling method to reduce information loss and provide a theoretical analysis of computation savings. The overall architecture is shown in Figure 2.

## 3.1 PRELIMINARY: ATTENTION IN VLM DECODERS

The VLM decoders typically use the causal *self-attention* in the original transformer model (Vaswani et al., 2017) for token interactions. Without loss of generality, we discuss the condition of single-head attention. Formally, the self-attention matrix with logits $\boldsymbol{A} \in \mathbb{R}^{L \times L}$, where $L$ denotes the length of total tokens (e.g., system, image, and question prompt tokens), is computed by

$$\boldsymbol{A} = \text{Attention}(\boldsymbol{Q}, \boldsymbol{K}) = \text{Softmax}\left(\frac{\boldsymbol{Q}\boldsymbol{K}^T}{\sqrt{D}}\right), \tag{1}$$

where the scalar $D$ represents the matrix dimension, and the $\boldsymbol{Q} \in \mathbb{R}^{L \times D}$ and $\boldsymbol{K} \in \mathbb{R}^{L \times D}$ are the query and key matrices, respectively. The keys and queries in a self-attention layer are computed in parallel by using multi-layer perceptrons (MLPs) to transform the input hidden states $\boldsymbol{H}$ into a common space, where aligned interactions between different modalities occur.

## 3.2 SPARSIFICATION GUIDANCE FROM TEXT TO VISUAL MODALITY

**Estimation of Visual Token Significance.** For a multimodal vision-language model, we should consider its impact on other modalities when deleting a single modal information. In our case, we need to understand how relevant a visual token is to the textual tokens in order to determine whether it should be removed. Therefore, we naturally come up with reusing the self-attention logits in VLMs transformer layers as a reference, since they already contain *language-to-vision* query results.

In particular, we take the interaction between the *query*-dimensional part of the textual modality and the *key*-dimensional part of the visual modality as the basis for sparsification priority matrix $\boldsymbol{P} \in \mathbb{R}^{L_t \times L_v}$, where $L_t$ and $L_v$ are the lengths of text and vision tokens, that is defined by

$$\boldsymbol{P} = \boldsymbol{A}[i_t, i_v], \tag{2}$$

with

$$i_t \in \{x | \boldsymbol{A}[x, :] = \mathbb{L}\}, \quad i_v \in \{y | \boldsymbol{A}[:, y] = \mathbb{I}\}, \tag{3}$$

where $\mathbb{L}$ and $\mathbb{I}$ denote the language instruction and image tokens set, respectively.

Next, we average scores of all instruction tokens to obtain the estimate $\bar{\boldsymbol{P}}_j$ for $j$th vision token as

$$\bar{\boldsymbol{P}}_j = \frac{1}{L_t} \sum_{i=1}^{L_t} \boldsymbol{P}[i, j], \quad j \in \{1, 2, \dots, L_v\}, \tag{4}$$

where we use $\bar{\boldsymbol{P}}_j$ as the significance indicator for sparsification and a larger value in $\bar{\boldsymbol{P}}_j$ means higher significance for the corresponding token. Calculation of equation 4 costs $L_t \times L_v$ FLOPs, while the correlation matrix $\boldsymbol{A}$ and the indexing process is *free*, which benefits the inference efficiency.

**Relevant Text Token Selection.** It is not appropriate to use all text tokens as a reference for visual sparsification. Figure 3 shows four representative cases where we compute the correlation between the prompt and the image. Case 3 highlights `Tylenol`, `Advil`, `ibuprofen`, while `top`, `sticker`, `fridge` in case 4 are significant, where a large proportion of question tokens in light red include little visual relevance. Therefore, it is unreasonable to make insignificant text tokens to rate vision tokens, and we need to select relevant text tokens (i.e., "*raters*") for guidance.

Specifically, for an input image $\boldsymbol{x}_v$, the vision embedding tokens $\boldsymbol{H}_v$ can be computed as

$$\boldsymbol{H}_v = \boldsymbol{W} \boldsymbol{Z}_v, \tag{5}$$

where $\boldsymbol{Z}_v$ is the visual feature provided by visual encoder $\boldsymbol{Z}_v = g(\boldsymbol{x}_v)$, and $\boldsymbol{W}$ is the projection matrix to convert $\boldsymbol{Z}_v$ into vision embedding tokens $\boldsymbol{H}_v$. For the language instruction $\boldsymbol{x}_q$, it is transformed into text embedding tokens $\boldsymbol{H}_q$ through the tokenizer. The above tokens both have the same dimensionality as the word embedding space. Then, we start to recognize which characters in the prompt are visually relevant and assign them the role of raters, which can be formulated as

$$\boldsymbol{S} = \{ i \mid \boldsymbol{R}[i] \geq m, \ i \in \{1, 2, \dots, L_t\}\}. \tag{6}$$

$$\boldsymbol{R} = \frac{1}{L_v} \sum_{j=1}^{L_v} (\text{Softmax}(\boldsymbol{H}_v \boldsymbol{H}_q{}^T))[j, :], \tag{7}$$

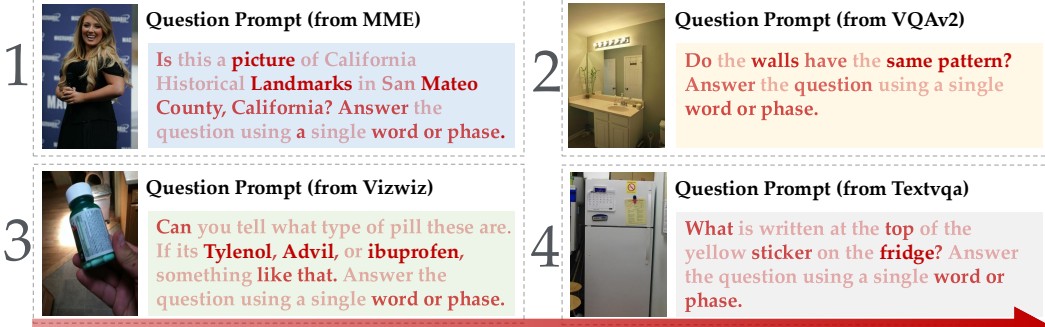

The level of text related to the image.

Figure 3: **Sample prompts from four representative multimodal benchmarks.** The darker the word, the greater its relationship to the image and the more valuable it is for reference. We see that some words are irrelevant to the vision domain (e.g., prepositions and pronouns) and should not be considered for visual sparsification. Best viewed in color.

where $m = \text{Mean}(\boldsymbol{R})$ and only candidates who exceed the $m$ will become raters. The strategy $\boldsymbol{S}$ defines the index of selected raters from the candidate list containing $L_t$ tokens with priority. The equation 7 costs $L_t \times L_v \times 2D$ FLOPs, only computed once before entering the decoder.

**Sparsification Level Adaptation.** Having obtained the token significance, we further propose a rank-based strategy to adaptively determine the level of vision sparsification at each decoder layer. Considering that *a full-rank matrix implies that all its rows or columns are linearly independent*, we use the rank of $\boldsymbol{P}$ to demonstrate the redundancy of the visual tokens. We argue that the difference between the dimension and rank of $\boldsymbol{P}$ reflects its redundancy and utilize a scaling factor $\lambda$ to determine the number of deletions as

$$N = \lambda \times (L_v - \text{Rank}(\boldsymbol{P})). \tag{8}$$

We then remove $N$ visual tokens with the smallest values in $\boldsymbol{P}$. Notably, if the result of $N$ in a decoder layer is 0, we skip the layer and abandon sparsification. This stage requires $L_t \times L_v \times \min(L_t, L_v)$ FLOPs for rank computation per layer.

## 3.3 VISUAL TOKEN RECYCLING

We progressively sparsify visual tokens in each layer in the decoder, which results in more discarded tokens at later stages. Despite less significant, the pruned vision tokens with relatively large values in $\boldsymbol{P}$ still contain certain information. To efficiently preserve more visual details with fewer tokens, we propose a token recycling strategy to aggregate and reconstruct tokens to be sparsified.

**Token Aggregation.** We first recycle the pruned visual tokens $\bar{h}_v$ with the top-$\tau$ (%) highest values in $\boldsymbol{P}$ from the deleted pool. Then, we group $\bar{h}_v$ tokens with $k$-nearest neighbor density peak aggregation algorithm (Rodriguez, 2014) for adaptive token aggregation.

In particular, we first compute the local density $\rho_i$ of the $i$th token of total $\tau \times N$ recycled tokens according to its $k$-nearest neighbors $\mathcal{K}(\bar{h}_v^i)$ as

$$\rho_i = \exp\left(-\frac{1}{k} \sum\nolimits_{\bar{h}_v^j \in \mathcal{K}(\bar{h}_v^i)}^{i,j} \left\| \bar{h}_v^i - \bar{h}_v^j \right\|_2^2 \right). \tag{9}$$

Then, we compute the minimum distance between the recycled token $\bar{h}_v^i$ and any other token with higher density (denoted as the distance indicator $\delta_i$) that is defined by

$$\delta_i = \begin{cases} \min & \left\| \bar{h}_v^i - \bar{h}_v^j \right\|_2, \text{ if } \exists j \text{ s.t. } \rho_j > \rho_i, \\ \max & \left\| \bar{h}_v^i - \bar{h}_v^j \right\|_2, \text{ otherwise }. \end{cases} \tag{10}$$

We use $\rho_i \times \delta_i$ to indicate the score of each token, where the tokens with higher scores are likely to be cluster centers. Other tokens are then assigned to the nearest cluster center via cosine similarity. The FLOPs cost in this stage is $L_r \times (3L_r - 1) \times 2D + L_r$, where $L_r = \tau \times N$ is the length of recycled tokens, $C = \theta \times L_r$ is the number of cluster centers, and $\tau$ and $\theta$ are hyperparameters.

**Token Reconstruction.** Having performed token aggregation, the recycled tokens with similar semantics are classified into the same group. Then, the tokens $\mathbb{T} \in \mathbb{R}^{N_k \times D}$ in the $k$th group are

reconstructed into a new compressed token $\boldsymbol{T}_k \in \mathbb{R}^{1 \times D}$ via the element-wise sum operation as

$$\boldsymbol{T}_k = \sum\nolimits_{i=1}^{N_k} \mathbb{T}[i], \ \ k \in \{1, 2, \ldots, C\}, \tag{11}$$

where $N_k$ is the token number of the $k$th group and the operation costs $D \times (L_r - C)$ FLOPs.

## 3.4 THEORETICAL ANALYSIS OF COMPUTATIONAL COMPLEXITY

We consider the computation of multi-head attention and feed-forward network (FFN) modules in the FLOPs estimation. Assuming $N$ is the number of pruned tokens, $D$ is the hidden state size, which is the same as the intermediate size in FFN, the FLOPs for one transformer layer can be reduced by $6(N-C)D^2 + 2(N-C)^2D$. Besides, our framework introduces additional computational overhead for the sparsification step with the details provided in Appendix A.2. Thus, we estimate the FLOP savings as the reduction part minus the additional sparsification overhead computed as

$$\underbrace{\sum\nolimits_i 6(N_i - C_i)D^2 + 2(N_i - C_i)^2 D}_{\text{reduction part}} - \underbrace{2L_t L_v D - \sum\nolimits_i L_t^i L_v^i (1 + \min(L_t^i, L_v^i)) - (6{L_r^i}^2 + 2L_r^i)D - L_r^i}_{\text{overhead part}}$$

$$\approx -2L_t L_v D + \sum\nolimits_i D(6DN_i(1-x) + N_i^2(2 + 2x^2 - 4x - 6(\tau)^2)) - {L_t^i}^2 L_v^i \tag{12}$$

$$\approx -2L_t L_v D + \sum\nolimits_i DN_i(6D + 2N_i) - {L_t^i}^2 L_v^i, \ \ i \in \{1, 2, \ldots, \Omega\},$$

where $\Omega$ is the number of total layers, and $x = \tau \times \theta$ is a very small decimal that can be ignored.

## 4 EXPERIMENTS

In this section, we validate our method within various VLM architectures on comprehensive multimodal benchmarks to assess its effectiveness including image and video understanding tasks.

### 4.1 IMAGE UNDERSTANDING TASKS

**Datasets.** For image-based multimodal evaluation, we conduct experiments on eight widely adopted benchmarks including GQA (Hudson & Manning, 2019), MMBench (MMB) (Liu et al., 2023b), MME (Fu et al., 2023), POPE (Li et al., 2023b), SQA (Lu et al., 2022), VQA$^{\text{V2}}$ (VQA V2) (Goyal et al., 2017), and VQA$^{\text{Text}}$ (TextVQA) (Singh et al., 2019). Furthermore, we check the consistency of SparseVLM on ConBench (Zhang et al., 2024b). More details are included in the Appendix A.4.

**Implementation Details.** We verify the proposed SparseVLM on two popular VLM frameworks: LLaVA (Liu et al., 2024) and Mini-Gemini (Li et al., 2024b). LLaVA-1.5 employs CLIP-pretrained ViT-L as the visual tower, while Mini-Gemini (MGM) further introduces a LAION-pretrained ConvNeXt-L (Liu et al., 2022) for high-resolution refinement. For LLaVA-1.5-7/13B and Mini-Gemini, we follow the same inference setting as the original paper as it is publicly available[1].

**Main Results.** In Table 1, we present the performance of SparseLLaVA (LLaVA equipped with SparseVLM) on image understanding benchmarks. To intuitively assess the performance, we provide the results by percentage format for comparative analysis, and the accuracy of the vanilla model with the 100% upper limit. We set 3 vision token count configurations (192, 128, and 64) to check the advantages of SparseVLM. When pruning from 576 to 192 tokens, the SparseLLaVA only decreases the average accuracy by $4.2\%$ without additional training and exceeds ToMe (Bolya et al., 2022) $7.4\%$. Furthermore, when only 64 tokens are left, our method outperforms FastV (Chen et al., 2024b) by a significant margin of **14.8%**, while ToMe performs worst due to its direct merging.

Figure 4 shows the performance of SparseMGM, and we visualize the results on POPE, TextVQA, and GQA. We find that our framework has an obvious advantage over FastV and ToMe. With the reduction of tokens, the gap between FastV and SparseVLM is increasing sharply. The reason is that, compared to FastV and ToMe, the text-aware strategy enables us to accurately locate visual tokens with more details, while the recycling of pruned tokens further reduces information loss.

Table 1: **Performance of SparseLLaVA under different vision token configurations.** The vanilla number of vision tokens is 576. The first line of each method is the raw accuracy of benchmarks, and the second line is the proportion relative to the upper limit. The last column is the average value.

| Method | GQA | MMB | MME | POPE | SQA | VQA$^{V2}$ | VQA$^{Text}$ | ConB | Avg. |
|---|---|---|---|---|---|---|---|---|---|
| *Upper Bound, 576 Tokens* **(100%)** | | | | | | | | | |
| Vanilla | 61.9 | 64.7 | 1862 | 85.9 | 69.5 | 78.5 | 58.2 | 19.8 | 100% |
| | 100% | 100% | 100% | 100% | 100% | 100% | 100% | 100% | |
| *Retain 192 Tokens* (↓ **66.7%**) | | | | | | | | | |
| ToMe (ICLR23) | 54.3 | 60.5 | 1563 | 72.4 | 65.2 | 68.0 | 52.1 | 17.4 | 88.4% |
| | 87.7% | 93.5% | 83.9% | 84.3% | 93.8% | 86.6% | 89.5% | 87.9% | |
| FastV (ECCV24) | 52.7 | 61.2 | 1612 | 64.8 | 67.3 | 67.1 | 52.5 | 18.0 | 88.1% |
| | 85.1% | 94.6% | 86.6% | 75.4% | 96.8% | 85.5% | 90.2% | 90.9% | |
| SparseVLM | 57.6 | 62.5 | 1721 | 83.6 | 69.1 | 75.6 | 56.1 | 18.8 | **95.8%** |
| | 93.1% | 96.6% | 92.4% | 97.3% | 99.4% | 96.3% | 96.4% | 94.9% | ↑ (**7.4%**) |
| *Retain 128 Tokens* (↓ **77.8%**) | | | | | | | | | |
| ToMe (ICLR23) | 52.4 | 53.3 | 1343 | 62.8 | 59.6 | 63.0 | 49.1 | 16.0 | 80.4% |
| | 84.7% | 82.4% | 72.1% | 73.1% | 85.8% | 80.2% | 84.4% | 80.8% | |
| FastV (ECCV24) | 49.6 | 56.1 | 1490 | 59.6 | 60.2 | 61.8 | 50.6 | 17.1 | 81.9% |
| | 80.1% | 86.7% | 80.0% | 69.4% | 86.6% | 78.7% | 86.9% | 86.4% | |
| SparseVLM | 56.0 | 60.0 | 1696 | 80.5 | 67.1 | 73.8 | 54.9 | 18.5 | **93.3%** |
| | 90.5% | 92.7% | 91.1% | 93.7% | 96.5% | 94.0% | 94.3% | 93.4% | ↑ (**11.4%**) |
| *Retain 64 Tokens* (↓ **88.9%**) | | | | | | | | | |
| ToMe (ICLR23) | 48.6 | 43.7 | 1138 | 52.5 | 50.0 | 57.1 | 45.3 | 14.0 | 70.2% |
| | 78.5% | 67.5% | 61.1% | 61.1% | 71.9% | 72.7% | 77.8% | 70.7% | |
| FastV (ECCV24) | 46.1 | 48.0 | 1256 | 48.0 | 51.1 | 55.0 | 47.8 | 15.6 | 72.1% |
| | 74.5% | 74.2% | 67.5% | 55.9% | 73.5% | 70.1% | 82.1% | 78.8% | |
| SparseVLM | 52.7 | 56.2 | 1505 | 75.1 | 62.2 | 68.2 | 51.8 | 17.7 | **86.9%** |
| | 85.1% | 86.9% | 80.8% | 87.4% | 89.4% | 86.9% | 89.0% | 89.4% | ↑ (**14.8%**) |

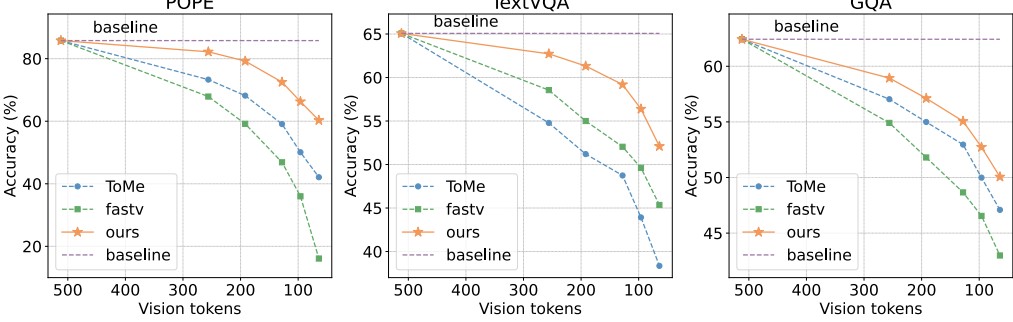

Figure 4: **Performance of MGM armed with SparseVLM on three multimodal benchmarks.** The horizontal axis represents the remaining number of vision tokens, while the vertical axis means the accuracy after percentage normalization. FastV is included for comparison.

## 4.2 VIDEO UNDERSTANDING TASKS

**Datasets.** We test our method on four common video question answering benchmarks, TGIF-QA (Jang et al., 2017), MSVD-QA (Xu et al., 2017), MSRVTT-QA (Xu et al., 2017) and ActivityNet-QA (Yu et al., 2019), where video-question pairs are massively disproportional in length. We adopt the

---

[1]github.com/haotian-liu/LLaVA and github.com/dvlab-research/MGM

Table 2: **The results of Video-LLaVA with SparseVLM on video question answering task.** The original number of video tokens is 2048, while our experiment collectively prunes it down to 135 tokens. FastV is included for comparison. The GPT-3.5 turbo is adopted for assistive evaluation.

| Method | TGIF | | MSVD | | MSRVTT | | ActivityNet | | Avg | |
|--------|------|-------|------|-------|--------|-------|-------------|-------|-----|-------|
| | Acc | Score | Acc | Score | Acc | Score | Acc | Score | Acc | Score |
| Video-LLaVA | 47.1 | 3.35 | 69.8 | 3.92 | 56.7 | 3.48 | 43.1 | 3.35 | 100.0% | +0.00 |
| FastV (ECCV24) | 23.1 | 2.47 | 38.0 | 2.71 | 19.3 | 2.02 | 30.6 | 2.82 | 52.1% | -1.02 |
| | 49.0% | -0.88 | 54.4% | -1.21 | 34.0% | -1.46 | 71.0% | -0.53 | | |
| SparseVLM | 44.7 | 3.29 | 68.2 | 3.90 | 31.0 | 2.68 | 42.6 | 3.32 | **86.5%** | **-0.17** |
| | 94.9% | -0.06 | 97.7% | -0.02 | 54.7% | -0.80 | 98.8% | -0.03 | ↑ (**34.4%**) | ↑ (**0.85**) |

evaluation framework proposed by Video-ChatGPT (Maaz et al., 2023) that utilizes both accuracy and ChatGPT score as key performance metrics with details in the Appendix A.4.

**Implementation Details.** We directly apply our SparseVLM for Video-LLaVA (Lin et al., 2023), which is a commonly used VLM framework for video question answering. Video-LLaVA is composed of several key components, including language bind encoder $f_M^v$ (Zhu et al., 2023a) for extracting features from raw visual inputs (e.g., images or videos), a language decoder model $f_L$ such as Vicuna (Touvron et al., 2023), a visual projection layer $f_P$, and a word embedding layer $f_T$. We adopt the same inference setup as the original Video-LLaVA code base[2], as it is publicly available.

**Main Results.** In Table 2, we set the Video-LLaVA with 2048 video tokens as our upper bound for an overall average accuracy of 100.0% and a score of +0.00. To make a fair comparison, we both preserve 135 vision tokens (93.4% pruning ratio) for FastV (Chen et al., 2024b) and SparseVLM. It is clear that our approach consistently outperforms FastV across all benchmarks, both in accuracy (Acc.) and GPT evaluation score. SparseVideoLLaVA achieves a total average accuracy of 86.5%, a significant **34.4%** higher than 52.1% of FastV. From the GPT score perspective, SparseVLM only loses 0.17 points compared to 1.02 points of FastV. These improvements suggest that when handling video modality containing temporal features, SparseVLM continues to deliver strong performance, generating accurate responses to diverse questions while utilizing significantly fewer tokens. This achieves an effective trade-off between inference efficiency and model performance.

## 5 ANALYSIS

### 5.1 EFFECTS OF RELEVANT TEXT TOKEN SELECTION

We propose a selection mechanism to localize visually irrelevant text tokens to limit their negative effects in rating the significance of vision tokens. Here we conduct experiments to analyze the effects of the mechanism in Figure 5. Under the same number of vision tokens (142), we have 3 settings (using all tokens, only text tokens, and only text raters we select) with LLaVA (Liu et al., 2023a) to judge vision token candidates. In TextVQA (Singh et al., 2019), our mechanism improves the vanilla text-aware method (only text tokens) by 0.79%, which validates that our extra selection is effective. Besides, by building upon the text-aware manner, we further outperform

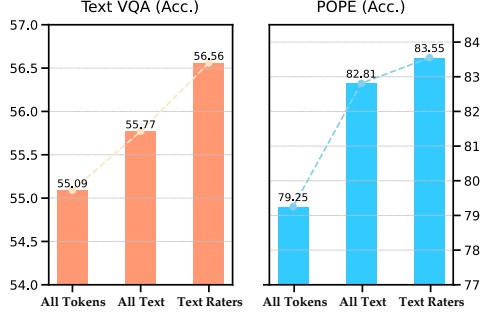

Figure 5: **The ablation study of text raters.**

the baseline (all tokens) by **4.3**% on POPE (Li et al., 2023b). The huge margin means POPE sparsification is quite sensitive to question prompts, and text guidance is necessary. In summary, text rater selection is general and improves the performance across various scenarios.

---

[2] github.com/PKU-YuanGroup/Video-LLaVA.

Table 3: **Ablation study on token reconstruction (TR).** Experiments are conducted on TextVQA and POPE using LLaVA with various sparsification ratios that highlight our TR generality.

| Benchmark | Tokens | | | | Avg |
|---|---|---|---|---|---|
| | 64 | 96 | 128 | 192 | |
| TextVQA | 49.6 | 52.9 | 54.2 | 55.7 | 53.1 |
| + TR | **51.6**(↑ 2.0) | **54.5**(↑ 1.6) | **55.0**(↑ 0.8) | **56.0**(↑ 0.3) | **54.3**(↑ 1.2) |
| POPE | 57.3 | 71.7 | 77.3 | 82.1 | 72.1 |
| + TR | **75.0**(↑ 17.7) | **78.2**(↑ 6.5) | **80.5**(↑ 3.2) | **83.6**(↑ 1.5) | **79.3**(↑ 7.2) |

Table 4: **Efficiency analysis of LLaVA with SparseVLM.** The detailed metric includes storage (cache memory), latency (CUDA time), and computation (FLOPs). $\Delta$ denotes the reduction ratio.

| Method | Token | Acc. | Storage Memory (MB) | $\Delta$ | CUDA Time (ms) ↓ | $\Delta$ | FLOPs (T) ↓ | $\Delta$ |
|---|---|---|---|---|---|---|---|---|
| Baseline | 576 | 100% | 302.4 | 100% | 419.9 | - | 9.6 | - |
| FastV | 192 | 88% | 100.8 | 66.7% | 290.8 | 30.7% | 2.3 | 76.0% |
| SparseVLM | 64 | 87% | **33.6** | 88.9% | **193.5** | 53.9% | **1.5** | 84.4% |

## 5.2 EFFECTS OF PRUNED TOKENS RECYCLING

To validate the effectiveness of our token recycling strategy, we perform ablation experiments on the LLaVA model (Liu et al., 2023a). The results are presented in Table 3. Across multiple sparsity ratios (64, 96, 128, 192), our algorithm achieves a significant average performance improvement of **1.2**% and **7.2**% on TextVQA (Singh et al., 2019) and POPE (Li et al., 2023b), respectively. Notably, as the number of pruned vision tokens increases, the benefit brought by our recycling method increases. For instance, when pruning from 192 to 64 tokens, the pruned token recycling significantly boosts the accuracy from **1.5%** to **17.7%** on POPE. We argue that when the size of the deleted pool grows, the amount of lost information increases. Our method effectively recycles the lost information and compresses it into few slots using the proposed reconstruction mechanism.

## 5.3 EFFICIENCY ANALYSIS

SparseVLM affords significant efficiency and storage gains for the inference process. We conduct a comparative analysis of storage memory, CUDA time, and FLOPs on LLaVA-7B, and compare our method with the baseline method and FastV (Chen et al., 2024b). As displayed in Table 4, we conduct an inference efficiency analysis on a single NVIDIA A100-80GB with identical lengths of text prompts and single-image inputs. Compared to the baseline model, SparseVLM achieves a significant reduction of 53.9% in CUDA time and 84.4% in FLOPs while keeping 88% accuracy. Despite SparseVLM has an additional overhead to calculate text raters and cluster-pruned vision tokens, it leads to fewer than FastV tokens with comparable accuracy. Additionally, SparseVLM demonstrates lower metrics in terms of CUDA latency time and FLOPs by 23.2% and 8.4%, respectively.

## 5.4 VISUALIZATION

As shown in Figure 6, we visualize SparseVLM on various VQA questions. From left to right, we visualize the results after we apply token pruning to different layers. On the very right, the dialogue box contains the prompt with the highlighted in red the most relevant text, and below is the generated answer in green produced by the remaining pruned image tokens. As the number of layers increases, more tokens are pruned and the Region of Interest (ROI) is gradually refined. The model systematically reduces less relevant image information while retaining key tokens that are closely tied to the question. The visualization reveals that SparseVLM, although discarding some overall image details, effectively retains essential visual tokens. These preserved tokens encapsulate

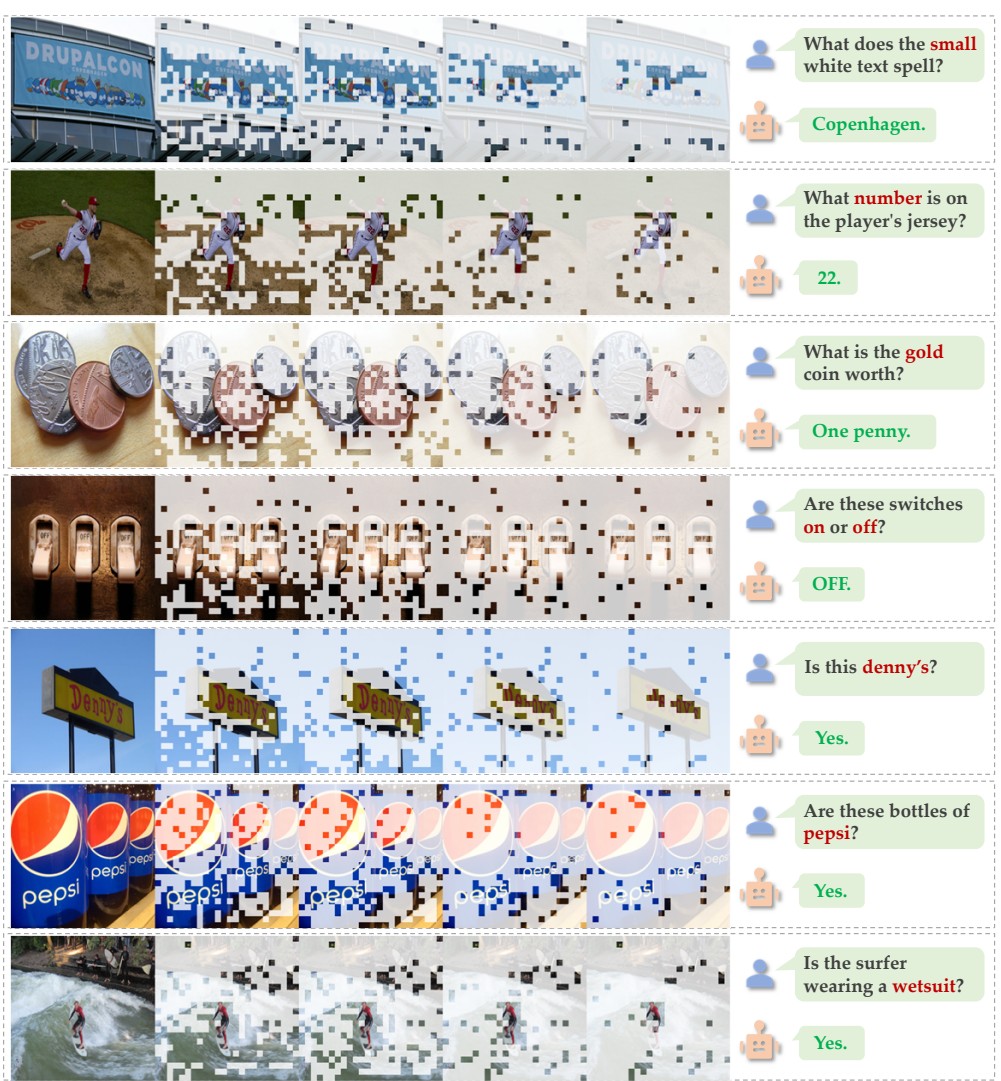

Figure 6: **Visualization of SparseVLM on different VQA prompts.** From left to right, the visual representation becomes increasingly sparse, leaving fewer vision tokens. Best viewed in color.

the features necessary for answering the question, focusing on more relevant visual regions through their interaction with the question. More visualization cases can be found in the Appendix A.6.

## 6 CONCLUSION

This paper introduced an efficient text-aware training-free token optimization mechanism called SparseVLM which significantly enhanced the efficiency of various VLMs in image and video understanding tasks. Unlike existing methods, SparseVLM was able to optimize VLMs without introducing extra parameters and fine-tuning costs. We achieved a more compact visual representation by progressively pruning the less relevant vision tokens. In addition, we employed the matrix rank to adaptively determine sparsification ratios and recycled the pruned tokens via reconstruction to reduce the information loss. Experiments demonstrated that SparseVLM increased the efficiency of various VLMs. Particularly, LLaVA equipped with SparseVLM achieved a reduction of 53.9% latency with a compression ratio of 88.9% while maintaining 87% accuracy. Moreover, our method exceeded FastV accuracy by 34.4% in video understanding tasks. Our SparseVLM can provide practical benefits for deploying off-the-shelf VLMs on edge devices and in the cloud.

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
