# A APPENDIX

## A.1 THE REDUNDANCY IN VISION TOKENS

In text-unrelated tasks, such as classification or segmentation, it is common to use a downsampling strategy which reduces redundancy in visual modality and makes the model more efficient to train (Zhang et al., 2024a). In Figure 7, which starts by comparing the original image with a downsampled version. The downsampled image reduces the number of tokens from 1166 to 576, achieving a 50% increase in efficiency. However, this process results in a 15% loss of information, as indicated by the decrease in entropy from 7.44 to 6.13. This trade-off is deemed acceptable for tasks unrelated to text such as classification or segmentation. For text-related tasks, such as visual question answering (VQA), there are two different modalities, text and vision. In this figure, the prompt is "What is written on the top of the yellow sticker on the fridge?" The output generated is "Warning". Pay attention to the highlighted part in both text and image, the text with the highest information density is highlighted with color, accounting for 88% of the total text; the region of interest (related to the prompt) part in the image only rates 38% in the whole image, which demonstrates that the information in images is typically more sparse than in natural language. Therefore, we proposed the SparseVLM to prune redundancy in visual tokens progressively. With our method, visual redundancy is reduced while maintaining the essential information required for accurate task performance, effectively improving the model's efficiency and effectiveness across different vision tasks.

## A.2 COMPUTING BUDGET DETAILED ESTIMATION

**Estimation of Visual Token Significance.** In this stage, only the equation 4 averaging process requires computation. Each vision token undergoes $L_t - 1$ additions and one division. With $L_v$ vision tokens in total, the number of FLOPs for this stage is $(L_t - 1 + 1) \times L_v = L_t \times L_v$.

**Relevant Text Selection.** In this process, given that official PyTorch implementation for Softmax and Averaging operations, the FLOPs for equation 7 can be approximately simplified to the matrix multiplication between $H_v$ and $H_q$. The result has a shape of $L_v \times L_t$, where each element undergoes $D$ multiplications and additions. Therefore, the FLOP count can be expressed as $L_t \times L_v \times 2D$.

**Sparsification Level Adaptation.** The rank of a matrix is typically computed using singular value decomposition (SVD) (Stewart, 1993). With the selected appropriate threshold, the number of above the threshold singular values determines the rank of the matrix. The FLOPs involved in this process can be approximated as $L_t \times L_v \times \min(L_t, L_v)$.

**Token Aggregation.** At this stage, the first part is to perform a nearest neighbor search for each element in the matrix. With the $L_r \times D$ matrix, this task can be simplified to calculate the distances between $L_r$ elements, resulting in a total of $L_r \times (L_r - 1)/2$ distance calculations. Each distance computation requires sequentially executing subtraction, squaring, addition, and square root

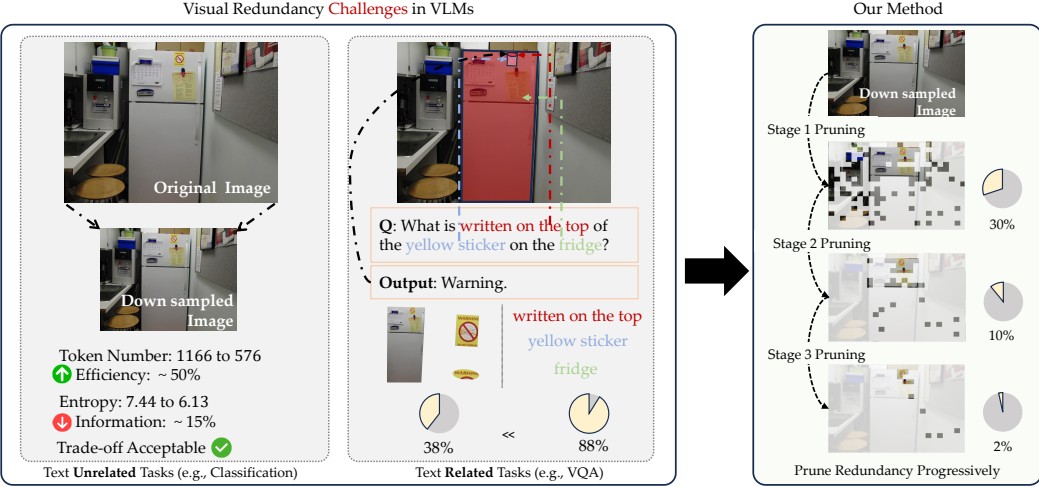

Figure 7: **Comparison of visual redundancy in different vision tasks.**

operations on $D$ elements. Consequently, the number of FLOPs in the nearest neighbor search is $L_r \times (L_r - 1)/2 \times 4D = L_r \times (L_r - 1) \times 2D$.

The second part is density calculation. Since the operations of averaging and applying the exponential function are implemented by the official PyTorch, this part can be simplified by the matrix squaring. Therefore, the FLOPs for this part are $L_r \times L_r \times 2D$.

The third part is distance indicator calculation. The computation can be approximately simplified to compute $\rho_i \times \delta_i$. Therefore, the FLOPs for this part can be approximated as $L_r \times L_r \times 2D$.

The last part is clustering. In this part, we need to select $C$ tokens with the highest scores from a total of $L_r$ tokens to serve as cluster centers, and the FLOPs can be approximated as $L$.

In summary, the total FLOPs for this stage are given by

$$\text{FLOPs} = \underbrace{L_r \times (L_r - 1) \times 2D}_{\text{Nearest Neighbors Search}} + \underbrace{L_r \times L_r \times 2D}_{\text{Density Calculation}} + \underbrace{L_r \times L_r \times 2D}_{\text{Distance Indicator Calculation}} + \underbrace{L}_{\text{Select Cluster Center}}$$

$$= L_r \times (3L_r - 1) \times 2D + L.$$

**Token Reconstruction.** Token reconstruction involves performing a weighted sum for each group, excluding the cluster center. Thus, there are $L_r - C$ elements to sum where each one has $1 \times D$ dimensions. Consequently, the number of FLOPs for this operation is $D \times (L_r - C)$.

## A.3 EFFICIENCY DETAILS

We present a comparative efficiency analysis of SparseVLM, the baseline, and FastV (Chen et al., 2024b) during the inference phase in Table 4. In this section, we provide additional details on the CUDA time measurement during the inference phase. Following VoCo-LLaMA Ye et al. (2024) setting, we primarily consider the following components that contribute to the reported CUDA time: image encoding time (if applicable), kv cache load time (if applicable), and transformers forward time. We exclude other computational times that are not dependent on the model itself and the caching strategy, such as model loading time, from the CUDA time measurement. Specifically, the attention operation is implemented by Sdpa Attention: `https://pytorch.org/tutorials/intermediate/scaled_dot_product_attention_tutorial`.

## A.4 DATASET

We conducted experiments on several widely used visual understanding benchmarks.

**GQA.** (Hudson & Manning, 2019) The GQA benchmark is composed of three parts: scene graphs, questions, and images. The image part contains images, as well as the spatial features of images and the features of all objects in images. The questions in GQA are designed to test the understanding of visual scenes and the ability to reason about different aspects of an image.

**MMBench.** (Liu et al., 2023b) The MMBench benchmark comprehensively evaluates the model's overall performance across multiple dimensions. It includes three levels of ability dimensions. The first level (L-1) consists of two main abilities, perception and reasoning. The second level (L-2) expands based on the first level, including six sub-abilities. The third level (L-3) further refines the second level, encompassing 20 specific ability dimensions. This hierarchical structure enables a granular and comprehensive evaluation of the model's various capabilities.

**MME.** (Fu et al., 2023) The MME benchmark is also a comprehensive benchmark meticulously designed to thoroughly evaluate various aspects of a model's performance. It consists of 14 subtasks that specifically aim to evaluate both the model's perceptual and cognitive abilities. By utilizing manually constructed instruction-answer pairs and concise instruction design, it effectively mitigates issues such as data leakage and unfair evaluation of model performance.

**POPE.** (Li et al., 2023b) The POPE benchmark is primarily used to evaluate the degree of Object Hallucination in models. It reformulates hallucination evaluation by requiring the model to answer a series of specific binary questions regarding the presence of objects in images. Accuracy, Recall, Precision, and F1 Score are effectively employed as reliable evaluation metrics to precisely measure the model's hallucination level under three different sampling strategies.

**ScienceQA.** (Lu et al., 2022) The ScienceQA benchmark covers a rich diversity of domains, including natural science, language science, and social science. Within each subject, questions are categorized first by the topic, then by the category, and finally by the skill. This hierarchical categorization results in 26 topics, 127 categories, and 379 skills, providing a comprehensive and diverse range of scientific questions. It provides a comprehensive evaluation of a model's capabilities in multimodal understanding, multi-step reasoning, and interpretability.

**VQA-v2.** (Goyal et al., 2017) The VQA-v2 benchmark evaluates the model's visual perception capabilities through open-ended questions. It consists of 265,016 images, covering a wide variety of real-world scenes and objects, providing rich visual contexts for the questions. For each question, there are 10 ground truth answers provided by human annotators, which allows for a comprehensive evaluation of the performance of different models in answering the questions accurately.

**TextVQA.** (Singh et al., 2019) The TextVQA benchmark focuses on the comprehensive integration of diverse text information within images. It meticulously evaluates the model's text understanding and reasoning abilities through a series of visual question-answering tasks with rich textual information. Models need to not only understand the visual content of the images but also be able to read and reason about the text within the images to answer the questions accurately.

**ConBench.** (Zhang et al., 2024b) The ConBench benchmark predominantly focuses on the consistency of the model's answers across a wide variety of different tasks and question types. It presents three core capabilities in a hierarchical manner, namely observation ability (sensation), complex reasoning (reasoning), and professional knowledge (knowledge). This hierarchical design aims to gradually challenge the performance of models on different tasks and provides fine-grained evaluation indicators, so as to evaluate the performance and consistency of the model.

**TGIF-QA.** (Jang et al., 2017) The TGIF-QA benchmark is an extension of the image question answering (ImageQA) task to the video domain, aiming to promote the development of video question answering techniques. It contains 165,000 question answer pairs in total and requires the model to comprehend the details of GIF videos. Specifically, it introduces three new tasks for VideoQA (repetition count, repeating action, and state transition), which require spatio-temporal reasoning from videos, and frame QA tasks that can be answered from one of the frames.

**MSVD-QA.** (Xu et al., 2017) The MSVD-QA benchmark is based on the existing Microsoft Research Video Description (MSVD) dataset and contains 1970 video clips and approximately 50.5K QA pairs. The questions and answers are diverse in nature, covering a wide range of topics and aspects related to the video content. Due to its relatively large data size and the diversity of questions, it is widely used for video question answering tasks and video caption tasks. The tasks formed in it are open-ended questions, consisting of five types of questions: what, who, how, when and where.

**MSRVTT-QA.** (Xu et al., 2017) The MSRVTT-QA benchmark consists of 10K video clips and 243k question answer pairs. One of the main challenges addressed by the MSRVTT-QA benchmark is the complexity of understanding and reasoning about video content. Videos contain both visual and temporal information, and models need to be able to effectively process and integrate these aspects to answer the questions accurately. The tasks formed in it also consist of five types of questions, similar to the MSVD-QA benchmark.

**ActivityNet-QA** (Yu et al., 2019) The ActivityNet-QA benchmark contains 58,000 human-annotated QA pairs on 5,800 videos derived from the ActivityNet dataset. The questions are designed to cover a range of types, including motion, spatial relationship, and temporal relationship, which challenge the model to understand and reason about the video content at different levels and evaluate the performance of VideoQA models in long-term spatio-temporal reasoning.

## A.5 IMPLEMENTATION DETAILS.

All of our experiments are conducted on a single Nvidia A100-80G GPU. The implementation was carried out in Python 3.10, utilizing PyTorch 2.1.2, CUDA 11.8, and transformers 4.31.0. The inference follows the evaluation settings established by LLaVA(Liu et al., 2024).

## A.6 VISUALIZATION

Figure 8 contains more visualization examples of SparseVLM on various VQA prompts.

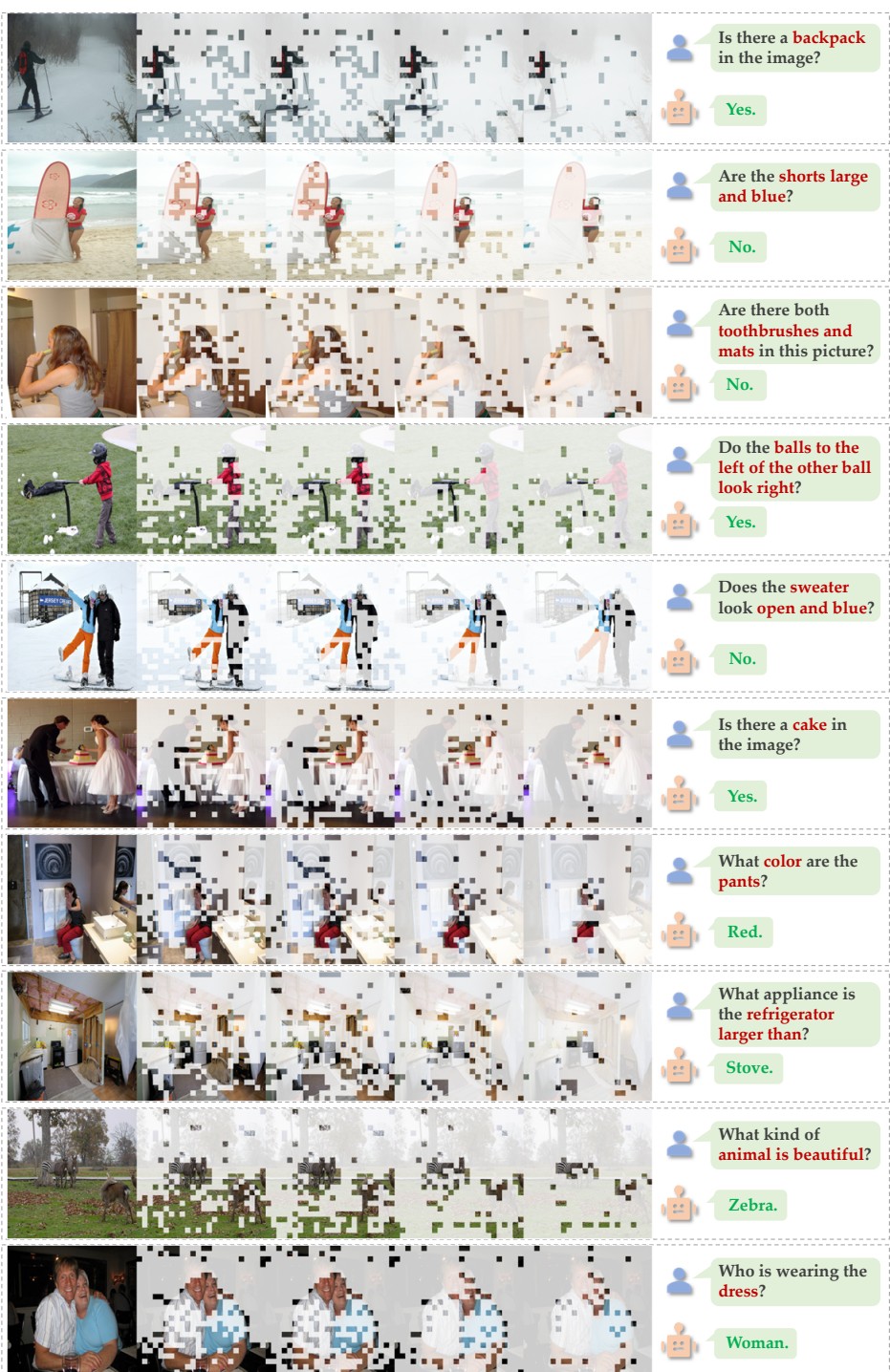

Figure 8: **More visualization examples of SparseVLM on different VQA prompts.**

## A.7 MORE DETAILED EFFICIENCY ANALYSIS

To better validate the efficiency of our method, we provide the latency-vs.-accuracy and FLOPs-vs.-Accuracy trade-offs for SparseVLM applied to LLaVA and MGM across three benchmarks: POPE, TextVQA, and MME, which are shown in Figure 9 and Figure 10. Besides, we also analyze Video-LLaVA matched with SparseVLM in Figure 11 on TGIF and MSVD.

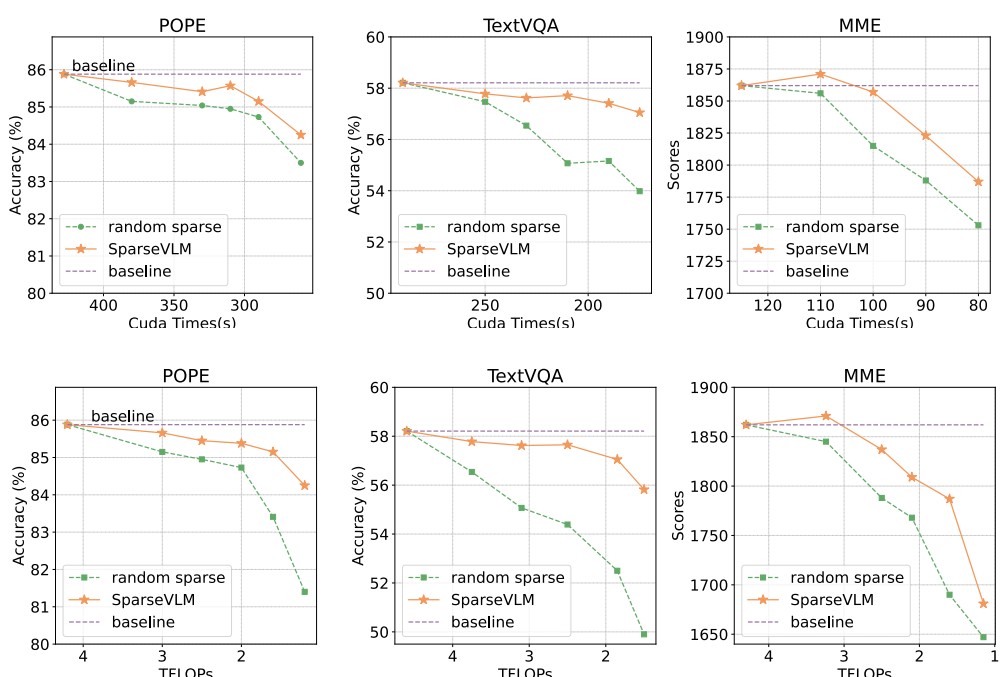

Figure 9: **Trade-offs for SparseVLM on LLaVA: (a) Latency vs. Accuracy, and (b) FLOPs vs. Accuracy. Both show comparisons of random sparse, SparseVLM, and baseline.**

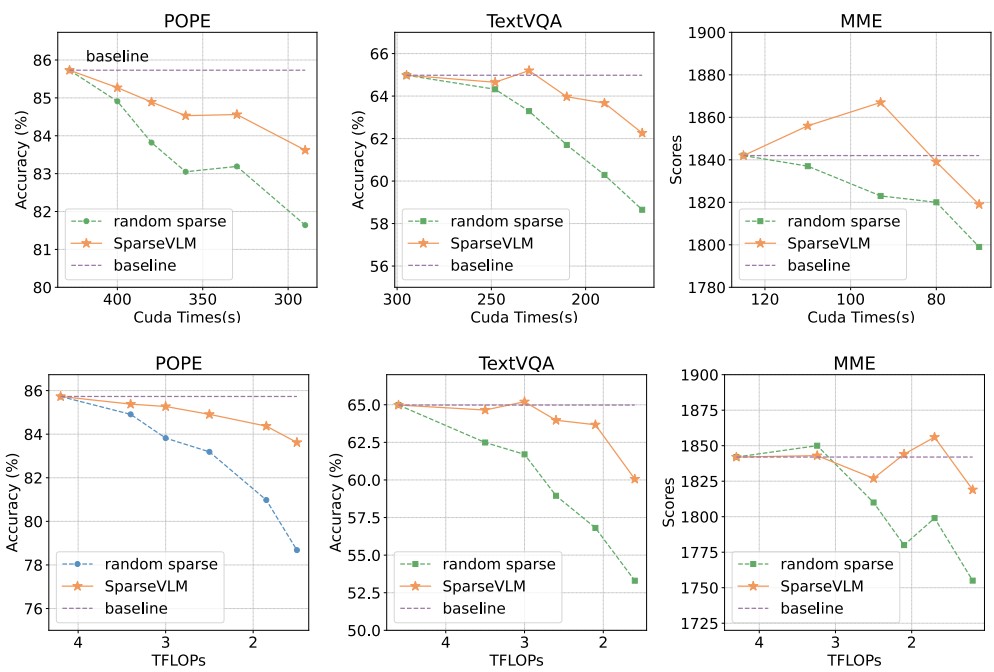

Figure 10: **Trade-offs for SparseVLM on MGM: (a) Latency vs. Accuracy, and (b) FLOPs vs. Accuracy. Both show comparisons of random sparse, SparseVLM, and baseline.**

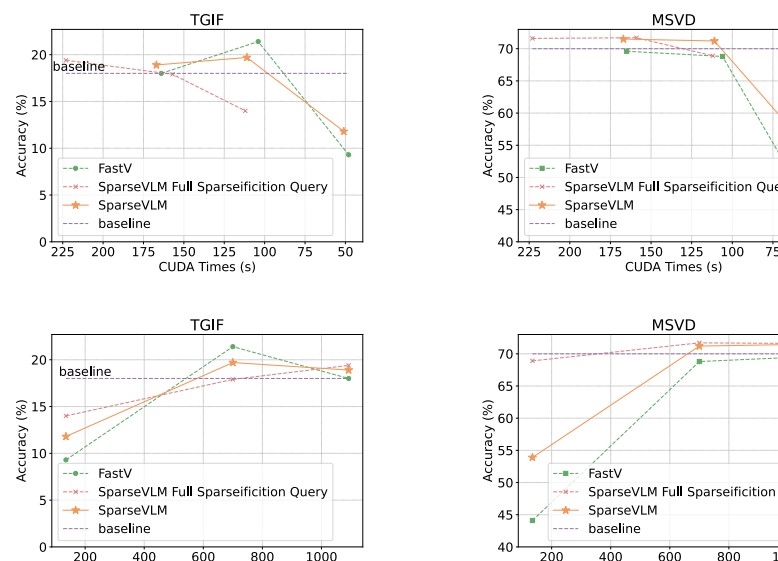

Figure 11: **Trade-offs for SparseVLM on Video-LLaVA: (a) Latency vs. Accuracy, and (b) Token budget vs. Accuracy. Both show comparisons of SparseVLM, FastV, and baseline.**

## A.8 COMPATIBILITY OF SPARSEVLM WITH FLASH ATTENTION

To ensure that SparseVLM remains compatible with Flash Attention even when pruning is applied, we devised a method to extract the mean value of the processed attention map without explicitly obtaining the full attention map. In decoder layers that do not require pruning, we use the original Flash Attention directly. For layers where pruning is necessary, we implemented an additional Flash Attention-based operation to directly obtain the mean attention scores w.r.t. the text raters, which is lightweight and also enjoys the efficiency of Flash Attention.

Specifically, the first forward pass operates identically to the original Flash Attention, generating the hidden states for all tokens before pruning. In the second forward pass, we introduce a specially designed $V$ matrix. In this matrix, for the rows corresponding to the text raters we wish to analyze, we set the values to the reciprocal of the number of text raters. This configuration allows the inner product between the attention map and the $V$ matrix to return the mean value of the attention scores for the selected text raters directly in Flash Attention.

Using this mean value, we perform a top-$k$ selection to identify the vision tokens to retain. Tokens that are excluded during this process are converted into masks, which are then applied to the hidden states produced by the first Flash Attention pass to complete the pruning operation. This method enables efficient integration of pruning with Flash Attention while preserving compatibility and computational efficiency.

### CORE PRINCIPLES AND CALCULATION OF SPARSEVLM FLASH ATTENTION

1. Attention Score Calculation

For each block $B$, compute the scaled dot-product attention scores:

$$S_B = \frac{Q_B K_B^T}{\sqrt{d_k}}$$

Here, $S_B$ is the attention score matrix computed within the block.

2. Block-wise Softmax

To ensure numerical stability, the softmax is computed in a stable manner using the log-sum-exp trick:

1. Subtract the maximum value for numerical stability:

$$S'_B = S_B - \max(S_B, \text{axis} = 1)$$

2. Normalize:

$$P_B = \frac{\exp(S'_B)}{\sum \exp(S'_B, \text{axis} = 1)}$$

3. Designation of $V$ Matrix

In order to return the mean value of the attention scores for the selected text raters directly in Flash Attention, we need to design a special $V$ matrix.

$$V_{ij} = \begin{cases} \frac{1}{n}, & \text{if } i \in \{i_1, i_2, \ldots, i_k\}, \\ \\ 0, & \text{otherwise.} \end{cases}$$

Here, $V$ is an $n \times d$ matrix, $n$ is the total number of rows in the matrix, $i$ is the row index, $1 \le i \le n$, $S = \{i \mid R[i] \ge m, i \in \{1, 2, \ldots, L_t\}\}$ define the text raters which we selected in Section 3.2.

4. Incremental Accumulation

Rather than storing $P$ explicitly, the result is directly accumulated into the output using:

$$O_B = P_B \cdot V_B$$

The final result is obtained by concatenating all blocks:

$$O = \text{Concat}(O_1, O_2, \ldots, O_B)$$

5. Streaming Softmax

When combining multiple blocks, an incremental softmax computation ensures that normalization is maintained across the entire sequence:

$$\text{softmax}(S) = \frac{\exp(S)}{\sum \exp(S)}$$

This avoids global dependencies and enables efficient block-wise computation.

6. Top-$k$ Selection for Vision Tokens

The top-$k$ selection can be expressed as:

$$O_k = \{x_i \in O_v \mid \text{rank}(x_i, O_v) \le k\},$$

$$O_v = \{y_j \in \text{mean}(O) \mid \text{vision tokens start} \le j \le \text{vision tokens end}\}.$$

where $O = \text{Concat}(O_1, O_2, \ldots, O_B)$ is the output array of the second Flash Attention, $O_v$ is the vision tokens part of $O$, $\text{rank}(x_i, O_v)$ represents the position of $x_i$ in $O_v$ when sorted in descending order.

The corresponding indices of the top-$k$ elements are:

$$I_k = \{i \mid x_i \in O_k\}.$$

7. Summary Formula for SparseVLM Flash Attention with Top-$k$ Selection

The complete process of SparseVLM Flash Attention can be summarized as:

$$I_k = \{i \mid x_i \in \{y_j \in O_v \mid \text{rank}(y_j, \text{mean}(\text{Concat}\left(\bigcup_B \text{softmax}\left(\frac{Q_B K_B^T}{\sqrt{d_k}} - \max(S_B)\right) \cdot V_B\right)$$
$$[\text{vision tokens start} : \text{vision tokens end}]))\}\}.$$

Here, each block $B$ is processed independently, and the results are combined using incremental normalization.