# OpenReview forum: "SparseVLM: Visual Token Sparsification for Efficient Vision Language Models Inference"
_ICLR.cc/2025/Conference — Submitted to ICLR 2025_

### Official Review · Reviewer_Rbrk · 2024-10-27

**Soundness:** 3
**Presentation:** 3
**Contribution:** 2
**Rating:** 3
**Confidence:** 4

**Summary:**

This paper introduces SparseVLM, an efficient, training-free optimization method for visual tokens in vision-language models (VLMs). Recognizing that visual tokens in VLMs often introduce high computational costs due to their low information density, SparseVLM selectively prunes redundant tokens without needing additional training data or parameters. By using the visual-relevant text tokens (from the self-attention matrix) to rate the importance of visual tokens, SparseVLM identifies and prunes unnecessary tokens progressively. A rank-based strategy is used to determine the pruning ratio per layer, while a token recycling method condenses pruned tokens into compact forms, maintaining essential visual information.

**Strengths:**

1. The paper presents SparseVLM, a training-free mechanism designed to improve the efficiency of vision-language models (VLMs) by optimizing the handling of visual tokens.
2. The paper is well-written and clearly presents the proposed framework. The authors provide detailed descriptions of their methodology.
3.  Considering the recycling of deleted image tokens is an effective method to alleviate performance degradation.

**Weaknesses:**

1. The primary focus on efficiency must maintain performance; otherwise, efficiency becomes meaningless. In Table 1, even the setting with the least acceleration, "Retain 192 Tokens," exhibits substantial performance drops across multiple benchmarks. Specifically, GQA drops by 4.3%, POPE by 2.3%, VQAv2 by 2.9%, MMB by 2.2%, and TextVQA by 2.1%, which are unacceptable losses.
2. In Section 5.1, why was the unusual number of 142 image tokens chosen for the experiment? Additionally, if the goal is to demonstrate the effectiveness of the "text rater," it would be insufficient to test only one efficiency setting. A range of settings retaining different proportions of image tokens should be used to substantiate its effectiveness across varying conditions.
3. In the section "Sparsification Level Adaptation",  N is calculated to determine the number of tokens deleted in each layer for adaptive purposes. However, in the later experimental sections, the number of retained image tokens (e.g., 192) is specified directly. If the result of N in a decoder layer is 0, how can you specify retained image tokens to 192? Isn’t this contradictory?
4. Rank(P) is a rather unusual way to compute visual redundancy. P represents a part of the attention map, but it is unclear why the linear correlation among attention vectors would relate to visual redundancy. Is there any supporting evidence for this, such as a reference to a paper?
5. Figure 1 shows that the patches selected by fastv are identical under different questions, which is unreasonable. Since fastv relies on the attention between text and image (this can be found in the source code), the selected patches should not be exactly the same. You may check for any errors in the process.
6. The paper mentions, "We reuse the self-attention matrix of visual-text tokens directly from the decoder layers without extra training parameters for sparsification." However, if the method requires outputting the self-attention matrix, it can not use FlashAttention, which would significantly impact inference speed.
7. In Table 1, it would be helpful to include efficiency evaluation like FLOPs and latency directly alongside performance scores on the benchmarks to facilitate comparison, the number of retained image tokens is not sufficient to evaluate efficiency.
8. One contribution claims, "it is the first attempt to explore the potential of text-aware guidance for efficient inference of VLMs." This is inaccurate, as the "fastv" approach also prunes image tokens based on text tokens’ attention to image tokens.
9. The description of the ToMe method in the Related Work section is inaccurate. "For example, ToMe (Bolya et al., 2022) prunes according to the relevance between visual tokens and text and merges both modalities through the BSM algorithm."
10. In the introduction, the calculation of the number of image tokens seems incorrect. The claim, "For instance, a 672 × 672 image in LLaVA (Liu et al., 2024) yields 2304 vision tokens that span over half of the context length," does not align with the correct calculation of 576 × 5 (four sub-images plus one resized original image). You can check it again, there might be an error somewhere.

**Questions:**

1. In Table 1, for the experimental results under the settings "Retain 192/128/64 Tokens," what exactly do these settings mean? For FastV, does this mean that only this number of image tokens is retained across all layers?
2. 5.1 section，"3 settings (using all tokens, only text tokens, and only text raters we select)"，explain the settings in detail.
3. Are you doing the pruning and recycling process in the prefilling stage? "we introduce a rank-based strategy to adaptively determine the sparsification ratio for each layer". If as said like this, do we prune and recycle at each layer in the prefilling stage to keep 192/128/64 tokens in experiment? Please give a clear explanation of your sparsification process, which is not stated in the paper.

---

> ### Author Response · Authors · 2024-11-22
> **Response to Reviewer Rbrk (Part 1 / 5)**
>
> We sincerely thank the reviewer Rbrk for the efforts in reviewing our paper.  Our responses according to the reviewer's comments are summarized as follows.
>
> -----------------
>
> > 1. In Table 1, even the setting with the least acceleration, "Retain 192 Tokens," exhibits substantial performance drops across multiple benchmarks.
>
> Thank you for your insightful comments.
>
> First, we want to clarify that there is an efficiency-performance trade-off in token sparsification. Our method achieves state-of-the-art performance at comparable efficiency levels with other methods. While "retain 192 tokens" is a more aggressive setting that may slightly impact performance, we can achieve marginal drops with more moderate sparsification ratios.
>
> As shown in the additional experiments in the table below, retaining mid-range token counts of 448, 384, and 320 allows us to maintain performance while realizing significant efficiency gains. For example, with 22% of the vision tokens deleted (resulting in 448 tokens), the average performance drop is only 0.5%.
>
> Note that this trade-off is adaptively controllable by our proposed sparsification level adaptation, which means that one can easily set the sparsification ratio to achieve the desired efficiency and performance.
>
> | Tokens | MME |  POPE | TextVQA
> | :--------  | :-----  | :----:  | :----:  |
> | 576 (original) | 1862| 85.9 | 58.2
> | 448 | 1845 (**99.1%**) |85.9 (**100%**) | 57.8 (**99.3%**)
> |384 | 1796 (**96.5%**) | 85.8 (**99.9%**) | 57.7 (**99.1%**)
> | 320 | 1778 (**95.5%**) | 85.2 (**99.2%**) | 57.6 (**99.0%**)
>
> -----------------
>
> > 2. Why was the unusual number of 142 image tokens chosen? Settings retaining different proportions of image tokens should be tested.
>
> (1) **Reason for 142 image tokens:** The 142 tokens are adaptively selected by the scaling factor $\lambda$ in equation (8), which is designed for fine-grained sparsifications in each layer.
>
> (2) **Experiments on various numbers of tokens remained:** We added experiments to test LLaVA with 320, 192, and 64 remaining tokens. As shown in the table below, our method with a rater selection mechanism obtains the optimal results, particularly in POPE (e.g., a gain of 2.2 is noted at the 192 tokens setting). Furthermore, we gain significant improvements compared to the approach in FastV [1] (using all tokens). This indicates that having more raters does not necessarily lead to better outcomes, while selecting raters that are relevant to the visual context is crucial.
>
> | Method|  POPE | TextVQA
> | :--------  | :-----  | :----:  |
> |**320 tokens**
> | + using all tokens | 79.7 | 56.8
> | + only text tokens |83.9 | 57.1
> | + only text raters  | **85.2** | **57.6**
> |**192 tokens**
> | + using all tokens | 75.8 | 54.4
> | + only text tokens |81.4 | 56.1
> | + only text raters | **83.6** | **56.7**
> | **64 tokens**
> | + using all tokens |64.2 | 49.8
> | + only text tokens | 71.9  | 51.3
> | + only text raters | **75.1** | **51.8**
>
> Through the above table, we observe that our rater selection mechanism is optimal, particularly obvious in the POPE  (e.g., a gain of 2.2 is noted at the 192 tokens setting). Furthermore, we verify that contrary to the approach of FastV [1], having more raters does not necessarily lead to better outcomes; rather, selecting raters that are relevant to the visual context is crucial.
>
> [1] Chen, Liang, et al. "An image is worth 1/2 tokens after layer 2: Plug-and-play inference acceleration for large vision-language models." European Conference on Computer Vision. Springer, Cham, 2025.
>
> -----------
>
> > 3. The number of tokens is adaptively determined by calculating N in the method, but in the experiments, the number is set to specific values (e.g., 192), how?
>
> (1) In fact, N directly determines the degree of sparsification, but the scaling factor $\lambda$ in equation (8) influences N. Therefore, we can delete a specific number of irrelevant vision tokens via $\lambda$. For instance, if you want to sparsify the number of vision tokens to 192 on MME, the $\lambda$ should be set to 13.5; if you wish to sparsify the number of vision tokens to 64 on TextVQA, the $\lambda$ should be set to 0.8.
>
> (2) The result of N in a decoder layer cannot be 0, because when the rank equals $L_v$, it indicates that P is full rank, and we will skip the sparsification for that layer as said in line 244.

---

> ### Author Response · Authors · 2024-11-23
> **Response to Reviewer Rbrk (Part 2 / 5)**
>
> -----------
>
> > 4. Why the linear correlation among attention vectors would relate to visual redundancy?
>
> We identified two relevant studies on the rank of the attention matrix. The first paper [1] shows a positive correlation between attention rank and Transformer performance, indicating that higher ranks lead to better model effectiveness until a saturation point is reached.
>
> The second paper [2] explores the limitations and redundancies of attention mechanisms, demonstrating that variations in attention matrix rank correlate strongly with visual feature redundancy.
>
> Building on these insights, we apply the concept of attention matrix rank to LLM decoder layers. The rank quantifies the linearly independent information within the matrix, reflecting relationships among tokens. Linearly dependent rows suggest redundancy, allowing us to retain one token while pruning others.
>
> We calculate attention matrix ranks for various tasks and adaptively prune tokens based on these ranks. Our extensive experiments show that attention matrices exhibit redundancy, enabling effective pruning without significant performance loss.
>
> [1] Min, Zeping, and Zhong Li. "On the Efficiency of Transformers: The Effect of Attention Rank."
>
> [2] OpenReview. "On the Limitation and Redundancy of Transformers: A Rank Perspective."
>
> ---
>
> > 5. Figure 1 shows that the patches selected by fastv are identical under different questions, which is unreasonable.
>
> (1) We sincerely apologize for this oversight. While the visualization in Figure 1 is correct, the caption contains an error and should be revised to 'VocoLLaMA [1].' Its sparsification is unrelated to the text.
>
> (2) Our method has a fundamental difference from FastV [2]. FastV does indeed evaluate vision tokens using text tokens, but it also incorporates the vision tokens themselves and the opinions of system tokens. Our approach, on the other hand, builds solely on text tokens and further filters the text raters to ensure their relevance to visual information. This effectiveness was also validated in Q2, where our method showed superiority.
>
> [1] Ye, Xubing, et al. "VoCo-LLaMA: Towards Vision Compression with Large Language Models." arXiv preprint arXiv:2406.12275 (2024).
>
> [2] Chen, Liang, et al. "An image is worth 1/2 tokens after layer 2: Plug-and-play inference acceleration for large vision-language models." European Conference on Computer Vision. Springer, Cham, 2025.
>
> ---
>
> > 6. SparseVLM compatibility with Flash attention and their comparison.
>
> Our SparseVLM can be compatible with FlashAttn. The following is our solution.
>
> ### **SparseVLM Flash Attention**:
>
> To ensure that SparseVLM remains compatible with Flash Attention, we devised a method to extract the mean value of the processed attention map without explicitly obtaining the full attention map. In normal decoder layers that do not require pruning, we use the original Flash Attention directly. For layers where pruning is necessary, we implemented an additional Flash Attention-based operation to directly obtain the mean attention scores w.r.t. the text raters, which is lightweight and also enjoys the efficiency in Flash Attention.
>
> Specifically, the first forward pass operates identically to the original Flash Attention, generating the hidden states for all tokens before pruning. In the second forward pass, we introduce a specially designed V matrix. In this matrix, for the rows corresponding to the text raters we wish to analyze, we set the values to $ 1 / \text {len(text raters)} $. This configuration allows the inner product between the attention map and the V matrix to return the mean value of the attention scores for the selected text raters directly in Flash Attention.
>
> Using this mean value, we perform a top-k selection to identify the vision tokens to retain. Tokens that are excluded during this process are converted into masks, which are then applied to the hidden states produced by the first Flash Attention pass to complete the pruning operation. This method enables efficient integration of pruning with Flash Attention while preserving compatibility and computational efficiency.
>
> ### **Core Principles and Calculation of SparseVLM Flash Attention**
>
> #### 1. Attention Score Calculation
>
> For each block $ B $, compute the scaled dot-product attention scores:
>
> $$
> S_B = \frac{Q_B K_B^T}{\sqrt{d_k}}
> $$
>
> Here, $ S_B $ is the attention score matrix computed within the block.
>
> #### 2. Block-wise Softmax
>
> To ensure numerical stability, the softmax is computed in a stable manner using the log-sum-exp trick:
>
> (1) Subtract the maximum value for numerical stability:
>
>    $$
>    S'_B = S_B - \max(S_B, \text{axis}=1)
>    $$
> (2) Normalize:
>
>    $$
>    P_B = \frac{\exp(S'_B)}{\sum \exp(S'_B, \text{axis}=1)}
>    $$

---

> > ### Author Response · Authors · 2024-11-28
> >
> > Dear reviewer Rbrk, we have updated the solution to the compatibility issue between Flash Attention and our method which you were particularly concerned about in the appendix. We are looking forward to your feedback!

---

> ### Author Response · Authors · 2024-11-23
> **Response to Reviewer Rbrk (Part 3 / 5)**
>
> #### 3. Designation of V Matrix
>
> In order to return the mean value of the attention scores for the selected text raters directly in Flash Attention, we need to design a special V matrix.
>
> $$
> V_{ij} =
> \begin{cases}
> \frac{1}{n}, & \text{if } i \in \\{i_1, i_2, \dots, i_k\\}, \\\\
> 0, & \text{otherwise}.
> \end{cases}
> $$
>
> Here, $ V $ is an $ n \times d $ matrix, $ n $ is the total number of rows in the matrix, $ i $ is the row index, $ 1 \leq i \leq n $, $ S = \\{ i \mid R[i] \geq m, \, i \in \\{1, 2, \dots, L_t\\} \\} $ define the text raters which we selected in Section 3.2.
>
> #### 4. Incremental Accumulation
>
> Rather than storing $ P $ explicitly, the result is directly accumulated into the output using:
>
> $$
> O_B = P_B \cdot V_B
> $$
>
> The final result is obtained by concatenating all blocks:
>
> $$
> O = \text{Concat}(O_1, O_2, \ldots, O_B)
> $$
>
> #### 5. Streaming Softmax
>
> When combining multiple blocks, an incremental softmax computation ensures that normalization is maintained across the entire sequence:
>
> $$
> \text{softmax}(S) = \frac{\exp(S)}{\sum \exp(S)}
> $$
>
> This avoids global dependencies and enables efficient block-wise computation.
>
> #### 6. Top-k selection for vision tokens
>
> The top-k selection can be expressed as:
>
> $$
> O_k = \\{ x_i \in O_v \mid \text{rank}(x_i, O_v) \leq k \\},
> $$
>
> $$
> O_v = \\{ y_j \in \text{mean}(O) \mid \text{vision tokens start} \leq j \leq \text{vision tokens end} \\}.
> $$
>
> where $ O = \text{Concat}(O_1, O_2, \ldots, O_B) $ is the output array of second Flash Attention, $ O_v $ is the vision tokens part of $ O $, $\text{rank}(x_i, O_v)$ represents the position of $x_i$ in $O_v$ when sorted in descending order.
>
> The corresponding indices of the top-k elements are:
>
> $$
> I_k = \\{ i \mid x_i \in O_k \\}.
> $$
>
> ### **Summary Formula**
>
> The complete process of SparseVLM Flash Attention can be summarized as:
>
> $$
> I_k = \\{ i \mid x_i \in \\{ y_j \in O_v \mid \text{rank}(y_j, \text{mean}(\text{Concat}\left( \bigcup_{B} \text{softmax}\left(\frac{Q_B K_B^T}{\sqrt{d_k}} - \max(S_B)\right) \cdot V_B \right) [\text{vtokens start} : \text{vtokens end}] )) \\} \\}
> $$
>
> Here, each block $ B $ is processed independently, and the results are combined using incremental normalization.
>
> [1] Dao, T. (2022). Flashattention: Fast and memory-efficient exact attention with io-awareness. Advances in Neural Information Processing Systems.
>
> [2] Dao, T. (2023). Flashattention-2: Faster attention with better parallelism and work partitioning. arXiv preprint arXiv:2307.08691.
>
> ### **Comparsion**
>
> Here, we utilized the mentioned SparseVLM FlashAttn in Part 2 compared with the baseline matched with FlashAttn. We conducted comprehensive experiments on LLaVA and MGM across three benchmarks: POPE, TextVQA, and MME. The results are shown in the following table. Besides, we compared our method with the random sparse matched with FlashAttn, and we observed that our method has significant improvement, when under a similar CUDA time.
>
>
> | Method                          | POPE (Acc) | CUDA Times (ms) | TextVQA (Acc) | CUDA Times (ms) | MME (Acc) | CUDA Times (ms)| Avg TFLOPs |
> |---------------------------------|----------|-----------------|-------------|--------------------|---------|----------------|------------|
> | Original LLaVA w/ Flash (576)   | 85.88    | 427696.68     | 58.21       | 286758.96       | 1862    | 125205.35    | 4.37       |
> | LLaVA (random Sparse w/ Flash)  | 84.67    | 314391.58     | 55.64       | 215478.40         | 1803    | 94158.56    | 2.25       |
> | **LLaVA (sparseVLM w/ Flash)**      | 85.21    | 315236.49     | 57.51       | 212753.22        | 1835    | 96313.14    | 2.24       |
> | Original MGM w/ Flash (576)     | 85.73    | 441471.83     | 64.98       | 294506.99        | 1842    | 129139.07    | 4.37       |
> | MGM (random Sparse w/ Flash)    | 83.32    | 351456.66     | 61.65       | 213259.19        | 1820    | 88876.37    | 2.40       |
> | **MGM (sparseVLM w/ Flash)**        | 84.57    | 351399.50     | 63.95       | 211810.73        | 1845    | 88883.89    | 2.39       |
>
> ---
>
> > 7. The additional efficiency experiments and explanation about latency and FLOPs on image understanding tasks.
>
> Thanks for the suggestion. We have complemented more experiments and revised the paper accordingly.
>
> (1) **More Trade-off Comparison**: Thank you for your detailed suggestions. We have included latency-vs.-accuracy and FLOPs-vs.-Accuracy tradeoffs for SparseVLM applied to LLaVA and MGM across three benchmarks: POPE, TextVQA, and MME. Figures 9 and 10 in Appendix A.7 now include a total of 12 sub-figures illustrating these tradeoffs on the image understanding tasks.
>
> (2) **Efficiency Summary**: The following table presents the latency-vs.-average accuracy and FLOPs-vs.-Average accuracy results for individual datasets, providing a comprehensive overview of our method's efficiency.

---

> ### Author Response · Authors · 2024-11-23
> **Response to Reviewer Rbrk (Part 4 / 5)**
>
> ### latency-vs.-average accuracy
>
> | Model                | POPE (Acc)    | CUDA Times (s)   | TextVQA (Acc)    | CUDA Times (s)   | MME (Acc)    | CUDA Times (s)   |
> |----------------------|--------|-------------|--------|-------------|--------|-------------|
> | LLaVA(sparseVLM)     | 85.2  | 315236.4944 | 57.5  | 212753.2193 | 1834.5 | 96313.14348 |
> | LLaVA(random Sparse) | 84.6  | 314391.5822 | 55.6  | 215478.4030  | 1803.0   | 94158.56141 |
> | MGM(sparseVLM)       | 84.7 | 351399.5041 | 64.0  | 211810.7318 | 1845.5| 88883.89499 |
> | MGM(random Sparse)   | 83.2 | 351456.6623 | 61.5  | 213259.1911 | 1819.5| 88876.37054 |
>
> ### TFLOPs-vs.-average accuracy
>
> | Model                | POPE (Acc)    | TFLOPs      | TextVQA (Acc)    | TFLOPs      | MME (Acc)    | TFLOPs      |
> |----------------------|--------|-------------|--------|-------------|--------|-------------|
> | LLaVA(sparseVLM)     | 85.8 | 2.081319069 | 57.4 | 2.531976786 | 1797.0   | 2.125052842 |
> | LLaVA(random Sparse) | 83.9 | 2.110299801 | 53.6  | 2.543639778 | 1747.6 | 2.124143099 |
> | MGM(sparseVLM)       | 84.7  | 2.460916468 | 63.5  | 2.561267015 | 1837.8 | 2.154662644 |
> | MGM(random Sparse)   | 82.3 | 2.47031752  | 58.6  | 2.57590639  | 1798.8 | 2.155692508 |
>
> In summary, the above experiments fully demonstrate the effectiveness of our method in reducing latency and computational complexity.
>
> ---
>
> > 8. The explanation of our claim "it is the first attempt to explore the potential of text-aware guidance for efficient inference of VLMs.
>
> Actually, our method indeed is the first work to explore the potential of text-aware guidance for the sparsification of VLMs. FastV [1] just simply utilizes all the tokens, including text tokens, vision tokens themselves, and system tokens to evaluate vision tokens. In contrast, our approach builds solely on text tokens and further filters the visual-aware text raters to improve performance. This effectiveness is also validated in Q2, where our method shows superiority.
>
> [1] Chen, Liang, et al. "An image is worth 1/2 tokens after layer 2: Plug-and-play inference acceleration for large vision-language models." European Conference on Computer Vision. Springer, Cham, 2025.
>
> > 9. The description of the ToMe method in the Related Work section is inaccurate.
>
> Thank you for your observation regarding the description of the ToMe [1] method in the Related Work section. We appreciate your attention to detail. We will revise the description to reflect the mechanism employed by ToMe accurately. Specifically, we have revised to "ToMe (Bolya et al., 2022) merge similar visual patches in Transformer blocks and speed up the match process through the Bipartite Soft Matching (BSM) algorithm".
>
> [1] Bolya D, Fu C Y, Dai X, et al. Token merging: Your vit but faster[J]. arXiv preprint arXiv:2210.09461, 2022.
>
> ---
>
> > 10. In the introduction, the calculation of the number of image tokens seems incorrect.
>
> Thank you for your careful review of related work. We acknowledge that the claim stating "a 672 × 672 image in LLaVA-HD yields 2304 vision tokens" is inappropriate. The accurate calculation should reflect that a 672 × 672 image results in 576 * 5 tokens, based on the configuration of four sub-images plus one resized original image, leading to a total of 2880 tokens. We corrected this in the revised manuscript and thank you for bringing this to our attention.
>
> ---
>
> > 11. In Table 1, for the experimental results under the settings "Retain 192/128/64 Tokens," what exactly do these settings mean? For FastV, does this mean that only this number of image tokens is retained across all layers?
>
> (1) **Clarification of Settings**: These settings are established to demonstrate the generality and robustness of our method under fewer tokens count. We select 64, 128, and 192 tokens at regular intervals (64) to evaluate performance across different token counts.
> (2) **Settings for FastV**: For FastV, we employ its pruning algorithm in the first layer, and this implies that only this specified number of image tokens is retained across all layers of the LLM decoder.

---

> ### Author Response · Authors · 2024-11-23
> **Response to Reviewer Rbrk (Part 5 / 5)**
>
> ---
>
> > 12. 5.1 section "3 settings (using all tokens, only text tokens, and only text raters we select)". explain the settings in detail.
>
> In this ablation experiment, we evaluate vision tokens with three different types of raters:
>
> (1) All tokens: this manner is the same as FastV, where all the tokens (text tokens, vision tokens, and the system token) function as raters. The pseudocode is as follows.
>
> ``relation_vis_text = self_attn_weights[:, :, v_token_start: v_token_start+v_token_num]``
>
> (2) All text tokens: this manner means only all the text tokens function as raters. The pseudocode is as follows.
>
> ``relation_vis_text = self_attn_weights[:, text_token_start:v_token_start, v_token_start: v_token_start + v_token_num]``
>
> (3) Text raters we select: based on the manner (2), we filter out the visual-relevant text tokens to evaluate vision tokens. The pseudocode is as follows.
>
> ``relation_vis_text = self_attn_weights[:, t_token_idx, v_token_start: v_token_start+v_token_num]``
>
> ---
>
> > 13. Please give a clear explanation of your sparsification process, which is not stated in the paper.
>
> (1) Yes, we do the pruning and recycling process during the prefilling stage, which occurs before the VLM begins its autoregressive token-by-token generation process.
>
> (2) In the prefilling stage, we start to prune and recycle in specific sparse layers (e.g., 2, 6, 15, 19), and have no operations in the left normal layers. During the generation process, we do not prune vision tokens anymore.  Therefore, the specific token numbers in experiments (e.g., 64, 128, 192) are computed in the prefilling stage.

---

> ### Author Response · Authors · 2024-11-25
> **Discussion to Reviewer Rbrk**
>
> We sincerely thank you for your efforts in reviewing our paper. We have provided corresponding responses and results, which we believe have covered your concerns. We hope to further discuss with you whether your concerns have been addresses or not. Please let us know if you still have any unclear part of our work.
>
> Best, Authors

---

> ### Author Response · Authors · 2024-12-02
> **Gentle Reminder Regarding Review of Reviewer Rbrk**
>
> **Dear reviewer Rbrk**,
>
> I hope this message finds you well.  We greatly appreciate the valuable feedback and suggestions you have provided so far. As the deadline approaches, we are eager to receive your feedback on our response and revisions. If possible, we kindly request an update on the progress of the review to ensure we can address any further comments or revisions promptly.
>
> Should you require any additional information or assistance from our end to help facilitate the review process, please do not hesitate to let us know. Your insights are highly valuable to us, and we genuinely appreciate your time and effort in reviewing our paper.
>
> Thank you for your patience and cooperation. We are looking forward to hearing from you soon.
>
> Warm regards,
>
> Submission402 Authors.

---

> ### Author Response · Authors · 2024-12-03
>
> Thank you for your response.
>
> Firstly, your rating of "5" in the previous review was determined by the initial 10 weaknesses, most of which we have addressed significantly. **Could you please specify any obvious issues that have emerged leading to a lower evaluation from you?** We are very open to receiving more detailed feedback from you, pinpointing the exact issues that remain unresolved. This feedback would greatly assist us in our future revisions.
>
> **Regarding the Raters used in FastV.** FastV works differently between with and without KV cache. We carefully examined the source code and found that it does indeed evaluate vision tokens using the last text token **only when the KV cache is applied**. Besides,  the statement in the text, 'compute the average attention score one token received from all other tokens as the criteria,' has proved it utilizes all the tokens.
>
> **The correction to FlashAttention.** In comparison to the baseline method Flash Attention, our approach demonstrates significant advantages in terms of FLOPs and CUDA Time. We are curious about how you arrived at "doubts about the fundamental understanding of this field.

---

### Official Review · Reviewer_7sbX · 2024-11-01

**Soundness:** 2
**Presentation:** 2
**Contribution:** 2
**Rating:** 5
**Confidence:** 5

**Summary:**

This paper proposes an efficient training-free token optimization mechanism dubbed SparseVLM without extra parameters or fine-tuning costs.

The contributions of this paper are summaried as follows:
1. The paper introduces a sparsification framework dubbed SparseVLM for vision-language models.
2. The paper first assigns visual-relevant text tokens as raters, adaptively prunes VLMs with the rank of the attention logits, and recycles partial tokens.
3. Consistently outperforms the FastV.

**Strengths:**

1. The paper is well-written, showcasing a clear and articulate presentation of ideas.
2. The paper is simple and easy to follow.
3. The training-free token optimization mechanism is more universal and can be better adapted to various VLM models compared to methods that require training.

**Weaknesses:**

1. One motivation of the paper is that visual tokens should be sparsified adaptively based on the question prompt. This prompt-aware sparsification, while preserving the original model's performance as much as possible, causes the VLM to lose its ability for multi-turn conversations.
2. The method in the paper requires explicitly obtaining the attention map, but in many inference acceleration frameworks, the attention map is not accessible, such as in FlashAttention. In Table 4, is the baseline using the standard attention mechanism? If compared with FlashAttention, does it still have a speed advantage?

**Questions:**

1. How to deal with RoPE for the sparsified visual tokens?
2. In Equation 7, why was it chosen to use the features from the visual encoder and text embeddings to select raters? Does this lead to the method performing poorly on problems that require logical reasoning, such as the performance on MMMU、Reasoning-related subset of MMBnech?

---

> ### Author Response · Authors · 2024-11-23
> **Response to Reviewer 7sbX (Part 1 / 3)**
>
> We sincerely thank the reviewer 7sbX for the efforts in reviewing our paper. Our responses according to the reviewer's comments are summarized as follows.
>
> ---
>
> > 1. Our SparseVLM performance on multi-turn conversations.
>
> Thanks for the comment. The compatibility with multi-turn dialogues is actually an advantage of our method over existing prompt-agnostic methods. The LLM in our method takes as inputs all the vision tokens and adaptively sparsify them according to the language prompt. When dealing with new questions, our SparseVLM can sparsify the vision tokens differently and is thus compatible with multi-turn dialogues. However, existing prompt-agnostic methods learn fixed visual compression regardless of the texts and cannot handle subsequent questions in multi-turn conversations.
>
> ---
>
> > 2. SparseVLM compatibility with Flash attention and their comparison.
>
> (1) **Clarification of FlashAttn Compatibility**: Our SparseVLM can be compatible with FlashAttn. The following is our solution.
>
> ### **SparseVLM Flash Attention**:
>
> To ensure that SparseVLM remains compatible with Flash Attention, we devised a method to extract the mean value of the processed attention map without explicitly obtaining the full attention map. In normal decoder layers that do not require pruning, we use the original Flash Attention directly. For layers where pruning is necessary, we implemented an additional Flash Attention-based operation to directly obtain the mean attention scores w.r.t. the text raters, which is lightweight and also enjoys the efficiency in Flash Attention.
>
> Specifically, the first forward pass operates identically to the original Flash Attention, generating the hidden states for all tokens before pruning. In the second forward pass, we introduce a specially designed V matrix. In this matrix, for the rows corresponding to the text raters we wish to analyze, we set the values to $ 1 / \text {len(text raters)} $. This configuration allows the inner product between the attention map and the V matrix to return the mean value of the attention scores for the selected text raters directly in Flash Attention.
>
> Using this mean value, we perform a top-k selection to identify the vision tokens to retain. Tokens that are excluded during this process are converted into masks, which are then applied to the hidden states produced by the first Flash Attention pass to complete the pruning operation. This method enables efficient integration of pruning with Flash Attention while preserving compatibility and computational efficiency.
>
> ### **Core Principles and Calculation of SparseVLM Flash Attention**
>
> #### 1. Attention Score Calculation
>
> For each block $ B $, compute the scaled dot-product attention scores:
>
> $$
> S_B = \frac{Q_B K_B^T}{\sqrt{d_k}}
> $$
>
> Here, $ S_B $ is the attention score matrix computed within the block.
>
> #### 2. Block-wise Softmax
>
> To ensure numerical stability, the softmax is computed in a stable manner using the log-sum-exp trick:
>
> (1) Subtract the maximum value for numerical stability:
>
>    $$
>    S'_B = S_B - \max(S_B, \text{axis}=1)
>    $$
> (2) Normalize:
>
>    $$
>    P_B = \frac{\exp(S'_B)}{\sum \exp(S'_B, \text{axis}=1)}
>    $$
>
> #### 3. Designation of V Matrix
>
> In order to return the mean value of the attention scores for the selected text raters directly in Flash Attention, we need to design a special V matrix.
>
> $$
> V_{ij} =
> \begin{cases}
> \frac{1}{n}, & \text{if } i \in \\{i_1, i_2, \dots, i_k\\}, \\\\
> 0, & \text{otherwise}.
> \end{cases}
> $$
>
> Here, $ V $ is an $ n \times d $ matrix, $ n $ is the total number of rows in the matrix, $ i $ is the row index, $ 1 \leq i \leq n $, $ S = \\{ i \mid R[i] \geq m, \, i \in \\{1, 2, \dots, L_t\\} \\} $ define the text raters which we selected in Section 3.2.
>
> #### 4. Incremental Accumulation
>
> Rather than storing $ P $ explicitly, the result is directly accumulated into the output using:
>
> $$
> O_B = P_B \cdot V_B
> $$
>
> The final result is obtained by concatenating all blocks:
>
> $$
> O = \text{Concat}(O_1, O_2, \ldots, O_B)
> $$
>
> #### 5. Streaming Softmax
>
> When combining multiple blocks, an incremental softmax computation ensures that normalization is maintained across the entire sequence:
>
> $$
> \text{softmax}(S) = \frac{\exp(S)}{\sum \exp(S)}
> $$
>
> This avoids global dependencies and enables efficient block-wise computation.
>
> #### 6. Top-k selection for vision tokens
>
> The top-k selection can be expressed as:
>
> $$
> O_k = \\{ x_i \in O_v \mid \text{rank}(x_i, O_v) \leq k \\},
> $$
>
> $$
> O_v = \\{ y_j \in \text{mean}(O) \mid \text{vision tokens start} \leq j \leq \text{vision tokens end} \\}.
> $$
>
> where $ O = \text{Concat}(O_1, O_2, \ldots, O_B) $ is the output array of second Flash Attention, $ O_v $ is the vision tokens part of $ O $, $\text{rank}(x_i, O_v)$ represents the position of $x_i$ in $O_v$ when sorted in descending order.
>
> The corresponding indices of the top-k elements are:
>
> $$
> I_k = \\{ i \mid x_i \in O_k \\}.
> $$

---

> > ### Author Response · Authors · 2024-11-28
> >
> > Dear reviewer 7sbX, we have updated the solution to the compatibility issue between Flash Attention and our method which you were particularly concerned about in the appendix. We are looking forward to your feedback!

---

> ### Author Response · Authors · 2024-11-23
> **Response to Reviewer 7sbX (Part 2 / 3)**
>
> ### **Summary Formula**
>
> The complete process of SparseVLM Flash Attention can be summarized as:
>
> $$
> I_k = \\{ i \mid x_i \in \\{ y_j \in O_v \mid \text{rank}(y_j, \text{mean}(\text{Concat}\left( \bigcup_{B} \text{softmax}\left(\frac{Q_B K_B^T}{\sqrt{d_k}} - \max(S_B)\right) \cdot V_B \right) [\text{vtokens start} : \text{vtokens end}] )) \\} \\}
> $$
>
> Here, each block $ B $ is processed independently, and the results are combined using incremental normalization.
>
> [1] Dao, T. (2022). Flashattention: Fast and memory-efficient exact attention with io-awareness. Advances in Neural Information Processing Systems.
>
> [2] Dao, T. (2023). Flashattention-2: Faster attention with better parallelism and work partitioning. arXiv preprint arXiv:2307.08691.
>
> (2) **Attention mechanism used in the baseline**: In our original implementation, we utilized the attention operation implemented by the official PyTorch, including our method and baseline.
>
> (3) **Speed experiments w & w/o FlashAttention**:
>
> As shown in the table below, we conducted speed experiments on LLaVA and MGM to compare SparseVLM with FlashAttention. Our method owns less latency while maintaining comparable accuracy.
> Besides, we compared our method with the random sparse matched with FlashAttention, and we observed that our method has significant improvement, when under a similar CUDA time.
> These findings, from both comparisons, demonstrate that SparseVLM not only achieves high accuracy but also provides an explicit speed advantage.
>
>
> | Method                          | POPE (Avg Acc) | Avg CUDA Times (ms) | TextVQA (Avg Acc) | Avg CUDA Times (ms) | MME (Avg Acc) | Avg CUDA Times (ms)| Avg TFLOPs |
> |---------------------------------|----------|-----------------|-------------|--------------------|---------|----------------|------------|
> | Original LLaVA w/ Flash (576)   | 85.88    | 427696.68     | 58.21       | 286758.96       | 1862    | 125205.35    | 4.37       |
> | LLaVA (random Sparse w/ Flash)  | 84.67    | 314391.58     | 55.64       | 215478.40         | 1803    | 94158.56    | 2.25       |
> | **LLaVA (sparseVLM w/ Flash)**      | 85.21    | 315236.49     | 57.51       | 212753.22        | 1835    | 96313.14    | 2.24       |
> | Original MGM w/ Flash (576)     | 85.73    | 441471.83     | 64.98       | 294506.99        | 1842    | 129139.07    | 4.37       |
> | MGM (random Sparse w/ Flash)    | 83.32    | 351456.66     | 61.65       | 213259.19        | 1820    | 88876.37    | 2.40       |
> | **MGM (sparseVLM w/ Flash)**        | 84.57    | 351399.50     | 63.95       | 211810.73        | 1845    | 88883.89    | 2.39       |
>
> ---
>
> >  3. How to deal with RoPE for the sparsified visual tokens?
>
> As is widely known, RoPE (Rotary Position Embedding) requires computation based on the kv_seq_len (key/value sequence length) and the position ID. In the original LLaVA, position IDs are assigned according to the sequence length of the current input to the decoder. In this setup, without token pruning, the position ID for each layer of the decoder remains fixed. Additionally, the key and value for each layer are stored in the KV cache, whose length matches the sequence length. During inference, after LLaVA generates the first token, it uses the KV cache to compute subsequent tokens by combining the current token with the previously stored key and value. Consequently, the position ID for each new token corresponds to the sequence length (with IDs starting from 0).
>
> However, in our SparseVLM setting, tokens are pruned, and the number of tokens retained in each layer of the KV cache can vary, with the length being less than or equal to the original sequence length. If we assign the position ID of a newly generated token based on the original sequence length, an error will occur during RoPE computation. This is because the position ID of the current token will not align with the pruned key/value sequence length retained in the KV cache.
>
> To address this issue, we propose a solution: we utilize LLaVA's KV cache to extract the length of the pruned key/value sequence retained in the current layer during the previous computation. Using this extracted length, we dynamically compute the position ID of the newly generated token. Subsequently, we calculate RoPE using the pruned key/value sequence length and the updated position ID. Whenever irrelevant tokens are removed or new tokens are inserted, our method reconstructs the positional encodings to ensure that the spatial relationships remain unchanged, regardless of the positional encoding method. This approach ensures compatibility between the position ID and the pruned sequence length, maintaining accurate computation and preserving the original positional relationships.

---

> ### Author Response · Authors · 2024-11-23
> **Response to Reviewer 7sbX (Part 3 / 3)**
>
> ----
>
> > 4. Why use the features from the visual encoder and text embeddings to select raters?
>
> Following the Pre-training stage of the VLM, the visual and textual modalities are effectively aligned, enabling direct similarity computations to select raters. Furthermore, we choose the selection before entering the LLM, rather than conducting it within the LLM sparse layers, to save computational resources.
>
> ----
>
> > 5. The performance of logical reasoning tasks.
>
> We conducted comprehensive experiments on four logical reasoning benchmarks, including MMMU, MMMU Pro, MMBench Attribute Reasoning (AR), and MMBench Logical Reasoning (LR). The results of our method are listed in the following table. We are surprised to observe that even with a severe reduction in the number of tokens from 576 to 192, the performance on MMMU and MMBench (AR) surpasses that of the baseline. Furthermore, in the 128-token setting, the average accuracy decreased by less than 2%, which serves to validate the efficacy of our approach. Notably, SparseLVM demonstrates superior performance in logical reasoning tasks compared to the overall results presented in Table 1. We attribute this enhancement to our method’s capability to eliminate extraneous redundant information, thereby enabling the LLM to concentrate more effectively on relevant visual information, which in turn enhances its logical reasoning abilities. Even in scenarios with limited visual information, such as the 64-token setting, our method still achieves not bad performance by aggregating useful information.
>
> | Method         | MMMU              | MMMU Pro         | MMBench (AR)    | MMBench (LR)    | Avg.      |
> | :-------------:| :----------------:| :---------------:| :--------------:| :--------------:| :--------: |
> | Upper Bound    | 34.8             | 30.3             | 73.3            | 30.5            | 100%      |
> | *192 tokens*   | 35.3 (**101.4%**) | 30.0 (99.0%)     | 71.9 (98.1%)    | 33.9 (**111.1%**) | **102.4%** |
> | *128 tokens*   | 34.9 (**100%**)   | 29.4 (97.0%)     | 70.4 (96.0%)    | 30.5 (**100%**) | 98.3%     |
> | *64 tokens*    | 32.1 (92.2%)      | 26.4 (87.1%)     | 67.2 (91.7%)    | 26.7 (87.5%)    | 89.6%     |

---

> ### Author Response · Authors · 2024-11-25
> **Discussion to Reviewer 7sbX**
>
> We sincerely thank you for your efforts in reviewing our paper. We have provided corresponding responses and results, which we believe have covered your concerns. We hope to further discuss with you whether your concerns have been addresses or not. Please let us know if you still have any unclear part of our work.
>
> Best, Authors

---

> ### Author Response · Authors · 2024-12-02
> **Gentle Reminder Regarding Review of Reviewer 7sbX**
>
> **Dear reviewer 7sbX**,
>
> I hope this message finds you well. We greatly appreciate the valuable feedback and suggestions you have provided so far. As the deadline approaches, we are eager to receive your feedback on our response and revisions. If possible, we kindly request an update on the progress of the review to ensure we can address any further comments or revisions promptly.
>
> Should you require any additional information or assistance from our end to help facilitate the review process, please do not hesitate to let us know. Your insights are highly valuable to us, and we genuinely appreciate your time and effort in reviewing our paper.
>
> Thank you for your patience and cooperation. We are looking forward to hearing from you soon.
>
> Warm regards,
>
> Submission402 Authors.

---

> > ### Comment · Reviewer_7sbX · 2024-12-02
> >
> > hank you to the authors for their efforts. Most of my concerns have been addressed. I will raise my rating to 5.

---

### Official Review · Reviewer_YHTW · 2024-11-03

**Soundness:** 3
**Presentation:** 3
**Contribution:** 3
**Rating:** 6
**Confidence:** 5

**Summary:**

This paper introduces SparseVLM, a training-free token optimization that improves the efficiency of Vision-Language Models (VLMs) by reducing visual token. The method improve the efficency of VLM by three steps: 1) first identify text tokens strongly correlated with visual signals via cross-attention, and then 2) measure the contribution of visual tokens to the selected visual-relevant text tokens (raters), and finally 3) adaptively prune the insignificant vision token. Experiments show the LLaVA equipped with SparseVLM reduces 61%∼67%
FLOPs with a compression ratio of 78% while maintaining 93% of the accuracy. The proposed method consistently outperforms the
existing state-of-the-art method FastV by 7.7%∼14.8% on LLaVA, 10.2%∼21.6% on MiniGemini, and 34.4% on VideoLLaVA.

**Strengths:**

1. The proposed Text-Guided Visual Token Pruning is novel, which  introduce text-aware guidance for visual token sparsification. The experiments showing this approach outperforms text-agnostic methods like FastV by 14.8% on LLaVA when retaining only 64 tokens, which validate the effectiveness of using text tokens as "raters" for visual importance.

2. The proposed method is training-free, which is easy to deploy.

3. The paper introduces a rank-based strategy to adaptively determine the sparsification ratio for each layer, which saves the number of hyperparameters and reduces the engineering effort.

4. Instead of directly pruning tokens, the proposed method merges them into compact representations. Ablation studies show this recycling mechanism improves accuracy from 1.5% to 17.7% on POPE when pruning to 64 tokens, demonstrating significant information preservation.

**Weaknesses:**

1. The proposed method requires the attention scores to select the visual tokens to be pruned. This would not be compatible with FlashAttention, and may require significantly more memory and possibly extra latency. The authors are encourage to do comparison with baselines powered with FlashAttention and show the result. My concerns is that without using FlashAttention the proposed method could cost much more memory, and make it harder or infeasible to be deployed. Specifically, author should show the peak memory consumption and latency comparision between proposed method vs Baseline with FlashAttention.

2. The experimental evaluation lacks comparison with latest token reduction methods that outperform FastV. Notably absent are Token summarization[https://arxiv.org/abs/2410.14072 ], Progressive Token Pruning[https://arxiv.org/abs/2301.13741]- all of which have better performance comparing to FastV in different tasks. Including these state-of-the-art baselines is essential for a comprehensive evaluation.

3. The experimental focuses on a single VLM architecture: LLaVA, which limiting evidence of the method's broader applicability. Testing across other VLM architectures like Flamingo would better demonstrate generalizability, particularly with different visual and textual feature fusion mechanisms.

**Questions:**

See weakness for details.

---

> ### Author Response · Authors · 2024-11-23
> **Response to Reviewer YHTW (Part 1 / 3)**
>
> We sincerely thank the reviewer YHTW for the efforts in reviewing our paper. Our responses according to the reviewer's comments are summarized as follows.
>
> ---
>
> > 1. SparseVLM compatibility with Flash attention and memory consumption issues.
>
> Thank you for your kind suggestion! To eliminate your concern about SparseVLM are not compatible with FlashAttention, we proposed an algorithm in detail below:
>
> ### **SparseVLM Flash Attention**:
>
> In normal decoder layers that do not require pruning, we use the original Flash Attention directly. For layers where pruning is necessary, we implemented an additional Flash Attention-based operation to directly obtain the mean attention scores w.r.t. the text raters, which is lightweight and also enjoys the efficiency in Flash Attention.
>
> Specifically, the first forward pass operates identically to the original Flash Attention, generating the hidden states for all tokens before pruning. In the second forward pass, we introduce a specially designed V matrix. In this matrix, for the rows corresponding to the text raters we wish to analyze, we set the values to $ 1 / \text {len(text raters)} $. This configuration allows the inner product between the attention map and the V matrix to return the mean value of the attention scores for the selected text raters directly in Flash Attention.
>
> Using this mean value, we perform a top-k selection to identify the vision tokens to retain. Tokens that are excluded during this process are converted into masks, which are then applied to the hidden states produced by the first Flash Attention pass to complete the pruning operation. This method enables efficient integration of pruning with Flash Attention while preserving compatibility and computational efficiency.
>
> ### **Core Principles and Calculation of SparseVLM Flash Attention**
>
> #### 1. Attention Score Calculation
>
> For each block $ B $, compute the scaled dot-product attention scores:
>
> $$
> S_B = \frac{Q_B K_B^T}{\sqrt{d_k}}
> $$
>
> Here, $ S_B $ is the attention score matrix computed within the block.
>
> #### 2. Block-wise Softmax
>
> To ensure numerical stability, the softmax is computed in a stable manner using the log-sum-exp trick:
>
> (1) Subtract the maximum value for numerical stability:
>
>    $$
>    S'_B = S_B - \max(S_B, \text{axis}=1)
>    $$
> (2) Normalize:
>
>    $$
>    P_B = \frac{\exp(S'_B)}{\sum \exp(S'_B, \text{axis}=1)}
>    $$
>
> #### 3. Designation of V Matrix
>
> In order to return the mean value of the attention scores for the selected text raters directly in Flash Attention, we need to design a special V matrix.
>
> $$
> V_{ij} =
> \begin{cases}
> \frac{1}{n}, & \text{if } i \in \\{i_1, i_2, \dots, i_k\\}, \\\\
> 0, & \text{otherwise}.
> \end{cases}
> $$
>
> Here, $ V $ is an $ n \times d $ matrix, $ n $ is the total number of rows in the matrix, $ i $ is the row index, $ 1 \leq i \leq n $, $ S = \\{ i \mid R[i] \geq m, \, i \in \\{1, 2, \dots, L_t\\} \\} $ define the text raters which we selected in Section 3.2.
>
> #### 4. Incremental Accumulation
>
> Rather than storing $ P $ explicitly, the result is directly accumulated into the output using:
>
> $$
> O_B = P_B \cdot V_B
> $$
>
> The final result is obtained by concatenating all blocks:
>
> $$
> O = \text{Concat}(O_1, O_2, \ldots, O_B)
> $$
>
> #### 5. Streaming Softmax
>
> When combining multiple blocks, an incremental softmax computation ensures that normalization is maintained across the entire sequence:
>
> $$
> \text{softmax}(S) = \frac{\exp(S)}{\sum \exp(S)}
> $$
>
> This avoids global dependencies and enables efficient block-wise computation.
>
> #### 6. Top-k selection for vision tokens
>
> The top-k selection can be expressed as:
>
> $$
> O_k = \\{ x_i \in O_v \mid \text{rank}(x_i, O_v) \leq k \\},
> $$
>
> $$
> O_v = \\{ y_j \in \text{mean}(O) \mid \text{vision tokens start} \leq j \leq \text{vision tokens end} \\}.
> $$
>
> where $ O = \text{Concat}(O_1, O_2, \ldots, O_B) $ is the output array of second Flash Attention, $ O_v $ is the vision tokens part of $ O $, $\text{rank}(x_i, O_v)$ represents the position of $x_i$ in $O_v$ when sorted in descending order.
>
> The corresponding indices of the top-k elements are:
>
> $$
> I_k = \\{ i \mid x_i \in O_k \\}.
> $$
>
> ### **Summary Formula**
>
> The complete process of SparseVLM Flash Attention can be summarized as:
>
> $$
> I_k = \\{ i \mid x_i \in \\{ y_j \in O_v \mid \text{rank}(y_j, \text{mean}(\text{Concat}\left( \bigcup_{B} \text{softmax}\left(\frac{Q_B K_B^T}{\sqrt{d_k}} - \max(S_B)\right) \cdot V_B \right) [\text{vtokens start} : \text{vtokens end}] )) \\} \\}
> $$
>
> Here, each block $ B $ is processed independently, and the results are combined using incremental normalization.
>
> [1] Dao, T. (2022). Flashattention: Fast and memory-efficient exact attention with io-awareness. Advances in Neural Information Processing Systems.
>
> [2] Dao, T. (2023). Flashattention-2: Faster attention with better parallelism and work partitioning. arXiv preprint arXiv:2307.08691.

---

> > ### Author Response · Authors · 2024-11-28
> >
> > Dear reviewer YHTW, we have updated the solution to the compatibility issue between Flash Attention and our method which you were particularly concerned about in the appendix. We are looking forward to your feedback!

---

> ### Author Response · Authors · 2024-11-23
> **Response to Reviewer YHTW (Part 2 / 3)**
>
> Here, we utilized the mentioned SparseVLM FlashAttn in Part 1 compared with the baseline matched with FlashAttn. We conducted comprehensive experiments on LLaVA and MGM across three benchmarks: POPE, TextVQA, and MME. The results are shown in the following table. Besides, we compared our method with the random sparse matched with FlashAttn, and we observed that our method has significant improvement, when under a similar CUDA time.
>
>
> | Method                          | POPE (Avg Acc) | Avg CUDA Times (ms) | TextVQA (Avg Acc) | Avg CUDA Times (ms) | MME (Avg Acc) | Avg CUDA Times (ms)| Avg TFLOPs |
> |---------------------------------|----------|-----------------|-------------|--------------------|---------|----------------|------------|
> | Original LLaVA w/ Flash (576)   | 85.88    | 427696.68     | 58.21       | 286758.96       | 1862    | 125205.35    | 4.37       |
> | LLaVA (random Sparse w/ Flash)  | 84.67    | 314391.58     | 55.64       | 215478.40         | 1803    | 94158.56    | 2.25       |
> | **LLaVA (sparseVLM w/ Flash)**      | 85.21    | 315236.49     | 57.51       | 212753.22        | 1835    | 96313.14    | 2.24       |
> | Original MGM w/ Flash (576)     | 85.73    | 441471.83     | 64.98       | 294506.99        | 1842    | 129139.07    | 4.37       |
> | MGM (random Sparse w/ Flash)    | 83.32    | 351456.66     | 61.65       | 213259.19        | 1820    | 88876.37    | 2.40       |
> | **MGM (sparseVLM w/ Flash)**        | 84.57    | 351399.50     | 63.95       | 211810.73        | 1845    | 88883.89    | 2.39       |
>
> ---
> ### **Memory consumption issues**
>
> The research in FastV shows that the attention allocation in the shallow layers is more balanced than that in the deeper layers [1]. Consequently, we do not perform sparsification on the first two layers. This leads to the fact that our experiments start sparsification from 576 tokens (taking LLaVA as an example). Therefore, peak memory consumption is similar to that of the approach without using the sparsification method. However, by sparsifying tokens, our method can significantly reduce the length of the KV Cache after the sparsified layers in the subsequent steps, thereby effectively reducing memory occupation and being suitable for running on devices such as mobile phones. Moreover, we calculate the memory occupied by vision tokens based on LLaVA, and the results are shown in the following table:
>
> |  Token |  ACC  | Storage Memory(MB) | Memory $\Delta$    |
> |  :-----: | :-----: | :------------------: | :-----: |
> |  576  | 100.0% |         302.4      |      0.0%   |
> |  192  | 95.8% |       100.8       | 66.7% |
> |  128  | 93.3% |        67.2        | 77.8% |
> |  64  | 86.9% |        33.6        | 88.9% |
>
> [1] Chen, Liang, et al. "An image is worth 1/2 tokens after layer 2: Plug-and-play inference acceleration for large vision-language models." European Conference on Computer Vision. Springer, Cham, 2025.
>
> ---
>
> > 2. The comparison of state-of-the-art baselines.
>
> Thank you for your valuable suggestion. Firstly, we need to clarify that our method is training-free, requiring no training resources. It only necessitates loading pre-trained weights for inference, which is simple and efficient.
>
> (1) Victor [1] was published on the arXiv platform in October 2024 and is concurrent with our work. In addition, Victor contains learnable parameters and is the same type of work as Voco-llama and llama-vid, while our SparseVLM is training-free, so it is unfair to compare it with Victor.
>
> (2) UPop [2] attains a high compression ratio, by progressively searching and retraining the subnet. However, the method needs to train trainable masks during the architecture search, which is inconsistent with our training-free SparseVLM. Therefore, it is unfair to compare it with UPop.
>
> Despite this, we still tried to find a state-of-the-art and training-free method PDrop [3] for comparison to demonstrate the superiority of our method, summarized as follows.
>
> Method| MME |  POPE | TextVQA
> |  :----:   |  :----:   | :----:  | :----:  |
> | 576 (original)| 1862| 85.9 | 58.2
> | 448 (PDrop) | 1601 | 84.8 | 57.5
> | **448 (SparseVLM)** | 1845|85.9 | 57.8
> | 128 (PDrop) | 1360 | 58.4 | 54.2
> | **128 (SparseVLM)** | 1490| 59.6 | 54.9
>
> [1] Wen, Y., Cao, Q., Fu, Q., Mehta, S., & Najibi, M. (2024). Efficient Vision-Language Models by Summarizing Visual Tokens into Compact Registers. arXiv preprint arXiv:2410.14072.
>
> [2] Shi, D., Tao, C., Jin, Y., Yang, Z., Yuan, C., & Wang, J. (2023, July). Upop: Unified and progressive pruning for compressing vision-language transformers. In International Conference on Machine Learning (pp. 31292-31311). PMLR.
>
> [3] Xing, L., Huang, Q., Dong, X., Lu, J., Zhang, P., Zang, Y., ... & Lin, D. (2024). Pyramiddrop: Accelerating your large vision-language models via pyramid visual redundancy reduction. arXiv preprint arXiv:2410.17247.

---

> ### Author Response · Authors · 2024-11-23
> **Response to Reviewer YHTW (Part 3 / 3)**
>
> ---
>
> > 3. The additional experiments about VLM architecture different from LLaVA.
>
> Thanks for your suggestion. Since the vanilla Flamingo is a prior work that is not as competitive as recent VLMs, we adopted our method on Qwen-VL on AI2D, TextVQA and MME to validate our generalizability across different architectures, as shown in the following tables:
>
> ### AI2D_TEST Performance Comparison
>
> | Tokens           | Accuracy | CUDA Times (ms)             | TFLOPs           |
> |------------------|----------|-------------------------|------------------|
> | 1365 (w/o sparse)| 79.27    | 350464.81      | 10.64 |
> | 768              | 77.23    | 290784.72       | 6.24  |
> | 512              | 76.13    | 244586.19       | 4.49 |
> | 256              | 73.25    | 200624.51      | 2.65 |
>
> ### TextVQA Performance Comparison
>
> | Tokens           | Accuracy | CUDA Times (ms)         | TFLOPs          |
> |------------------|----------|-------------------------|------------------|
> | 1326 (w/o sparse)| 84.30     | 524723.32       | 10.17 |
> | 768              | 80.86    | 462638.02       | 6.08 |
> | 512              | 79.96    | 371997.75      | 4.27  |
> | 256              | 73.47    | 312265.60       | 2.48  |
>
> ### MME Performance Comparison
>
> | Tokens           | Accuracy | CUDA Times (ms)         | TFLOPs           |
> |------------------|----------|-------------------------|-------------------|
> | 1315 (w/o sparse)| 2305     | 263030.24       | 10.04 |
> | 768              | 2184     | 216253.95      | 6.11  |
> | 512              | 2175     | 173129.89       | 4.28  |
> | 256              | 2167     | 145739.92       | 2.49 |
>
> In summary, our approach achieved strong performance on the Qwen-VL architecture. Even after pruning approximately 42% of the tokens on the TextVQA, MME, and AI2D benchmarks, we retained at least 95% accuracy of the original accuracy. Additionally, our method demonstrated significant computational efficiency, reducing CUDA times and TFLOPs by approximately 17.8% to 41.4%.

---

> ### Author Response · Authors · 2024-11-25
> **Discussion to Reviewer YHTW**
>
> We sincerely thank you for your efforts in reviewing our paper. We have provided corresponding responses and results, which we believe have covered your concerns. We hope to further discuss with you whether your concerns have been addresses or not. Please let us know if you still have any unclear part of our work.
>
> Best, Authors

---

> ### Author Response · Authors · 2024-12-02
> **Gentle Reminder Regarding Review of Reviewer YHTW**
>
> **Dear reviewer YHTW**,
>
> I hope this message finds you well. We greatly appreciate the valuable feedback and suggestions you have provided so far. As the deadline approaches, we are eager to receive your feedback on our response and revisions. If possible, we kindly request an update on the progress of the review to ensure we can address any further comments or revisions promptly.
>
> Should you require any additional information or assistance from our end to help facilitate the review process, please do not hesitate to let us know. Your insights are highly valuable to us, and we genuinely appreciate your time and effort in reviewing our paper.
>
> Thank you for your patience and cooperation. We are looking forward to hearing from you soon.
>
> Warm regards,
>
> Submission402 Authors.

---

> > ### Comment · Reviewer_YHTW · 2024-12-02
> >
> > The authors addressed some of my concerns. I intend to maintain my rating.

---

### Official Review · Reviewer_hiZv · 2024-11-04

**Soundness:** 2
**Presentation:** 3
**Contribution:** 2
**Rating:** 6
**Confidence:** 5

**Summary:**

This paper introduces SparseLVM, a training-free method to prune redundant visual tokens in LVLMs. SparseLVM leverages visual-relevant text tokens to rate the significance of vision tokens within the self-attention matrix, leading to the progressive pruning of irrelevant visual tokens. Specifically, SparseLVM proposes a rank-based strategy to adaptively determine the sparsification ratio for each layer and a token recycling method that compresses pruned tokens into center tokens. SparseLVM reduces the number of tokens with less performance drop than ToMe and FastV.

**Strengths:**

1. The proposed SparseLVM framework, including rank-based strategy and token recycling, is reasonable.

2. The paper is clear to read. It is easy for the audience to follow the sophisticated designs in SparseLVM.

3. Experiments are performed on both image and video benchmarks.

**Weaknesses:**

1. The proposed SparseLVM is not practical for two reasons.

- First, it is not compatible with FlashAttn, which is a standard solution for accelerating the calculation of self-attention. In SparseLVM, the attention matrix must be explicitly obtained to select redundant visual tokens in **each layer** of LVLMs. However, FlashAttn does not support attaining the explicit attention matrix. Without compatibility with FlashAttn, SparseLVM will be limited in its efficiency. The SparseLVM should be compared with the original LVLMs with FlashAttn.
- Second, although the performance drop of SparseLVM is less than ToMe and FastV, it is still considerably large. More explanations and discussions are necessary.

2. Some important ablation studies are not shown.

- For verifying efficiency, the SparseLVM should be compared with the original LVLMs with FlashAttn.
- For verifying effectiveness, the SparseLVM should report more results on high-resolution image understanding benchmarks, such as DocVQA, InfoVQA, AI2D, etc, as in leading LVLMs [1].


3. Some details of SparseLVM are not clearly introduced.
- What is the value of m in equation (6), lambda in equation (8), and tau in equation (9)? How does SparseLVM determine them?
- After Visual Token Recycling, how does SparseLVM insert these recycled tokens into the preserved tokens? It seems that these recycled tokens have the risk of spatial relationship between different image tokens.



[1] Qwen2-VL: Enhancing Vision-Language Model’s Perception of the World at Any Resolution

**Questions:**

Due to the weakness of this paper, I tend to be borderline negative about this paper. See weakness section for details of my concerns and questions.



###################

Thanks for your reply. Most of my concerns are solved. I will raise my rating to 6. Hope to see the SparseVLM with Flash Attention to be released.

---

> ### Author Response · Authors · 2024-11-23
> **Response to Reviewer hiZv (Part 1 / 3)**
>
> We sincerely thank the reviewer hiZv for the efforts in reviewing our paper. Our responses according to the reviewer's comments are summarized as follows.
>
> ---
>
> > 1. Clarification of FlashAttn Compatibility
>
> Our SparseVLM can be compatible with FlashAttn. The following is our solution:
>
> ### **SparseVLM Flash Attention**:
>
> To ensure that SparseVLM remains compatible with Flash Attention, we devised a method to extract the mean value of the processed attention map without explicitly obtaining the full attention map. In normal decoder layers that do not require pruning, we use the original Flash Attention directly. For layers where pruning is necessary, we implemented an additional Flash Attention-based operation to directly obtain the mean attention scores w.r.t. the text raters, which is lightweight and also enjoys the efficiency in Flash Attention.
>
> Specifically, the first forward pass operates identically to the original Flash Attention, generating the hidden states for all tokens before pruning. In the second forward pass, we introduce a specially designed V matrix. In this matrix, for the rows corresponding to the text raters we wish to analyze, we set the values to $ 1 / \text {len(text raters)} $. This configuration allows the inner product between the attention map and the V matrix to return the mean value of the attention scores for the selected text raters directly in Flash Attention.
>
> Using this mean value, we perform a top-k selection to identify the vision tokens to retain. Tokens that are excluded during this process are converted into masks, which are then applied to the hidden states produced by the first Flash Attention pass to complete the pruning operation. This method enables efficient integration of pruning with Flash Attention while preserving compatibility and computational efficiency.
>
> ### **Core Principles and Calculation of SparseVLM Flash Attention**
>
> #### 1. Attention Score Calculation
>
> For each block $ B $, compute the scaled dot-product attention scores:
>
> $$
> S_B = \frac{Q_B K_B^T}{\sqrt{d_k}}
> $$
>
> Here, $ S_B $ is the attention score matrix computed within the block.
>
> #### 2. Block-wise Softmax
>
> To ensure numerical stability, the softmax is computed in a stable manner using the log-sum-exp trick:
>
> (1) Subtract the maximum value for numerical stability:
>
>    $$
>    S'_B = S_B - \max(S_B, \text{axis}=1)
>    $$
> (2) Normalize:
>
>    $$
>    P_B = \frac{\exp(S'_B)}{\sum \exp(S'_B, \text{axis}=1)}
>    $$
>
> #### 3. Designation of V Matrix
>
> In order to return the mean value of the attention scores for the selected text raters directly in Flash Attention, we need to design a special V matrix.
>
> $$
> V_{ij} =
> \begin{cases}
> \frac{1}{n}, & \text{if } i \in \\{i_1, i_2, \dots, i_k\\}, \\\\
> 0, & \text{otherwise}.
> \end{cases}
> $$
>
> Here, $ V $ is an $ n \times d $ matrix, $ n $ is the total number of rows in the matrix, $ i $ is the row index, $ 1 \leq i \leq n $, $ S = \\{ i \mid R[i] \geq m, \, i \in \\{1, 2, \dots, L_t\\} \\} $ define the text raters which we selected in Section 3.2.
>
> #### 4. Incremental Accumulation
>
> Rather than storing $ P $ explicitly, the result is directly accumulated into the output using:
>
> $$
> O_B = P_B \cdot V_B
> $$
>
> The final result is obtained by concatenating all blocks:
>
> $$
> O = \text{Concat}(O_1, O_2, \ldots, O_B)
> $$
>
> #### 5. Streaming Softmax
>
> When combining multiple blocks, an incremental softmax computation ensures that normalization is maintained across the entire sequence:
>
> $$
> \text{softmax}(S) = \frac{\exp(S)}{\sum \exp(S)}
> $$
>
> This avoids global dependencies and enables efficient block-wise computation.
>
> #### 6. Top-k selection for vision tokens
>
> The top-k selection can be expressed as:
>
> $$
> O_k = \\{ x_i \in O_v \mid \text{rank}(x_i, O_v) \leq k \\},
> $$
>
> $$
> O_v = \\{ y_j \in \text{mean}(O) \mid \text{vision tokens start} \leq j \leq \text{vision tokens end} \\}.
> $$
>
> where $ O = \text{Concat}(O_1, O_2, \ldots, O_B) $ is the output array of second Flash Attention, $ O_v $ is the vision tokens part of $ O $, $\text{rank}(x_i, O_v)$ represents the position of $x_i$ in $O_v$ when sorted in descending order.
>
> The corresponding indices of the top-k elements are:
>
> $$
> I_k = \\{ i \mid x_i \in O_k \\}.
> $$
>
> ### **Summary Formula**
>
> The complete process of SparseVLM Flash Attention can be summarized as:
>
> $$
> I_k = \\{ i \mid x_i \in \\{ y_j \in O_v \mid \text{rank}(y_j, \text{mean}(\text{Concat}\left( \bigcup_{B} \text{softmax}\left(\frac{Q_B K_B^T}{\sqrt{d_k}} - \max(S_B)\right) \cdot V_B \right) [\text{vtokens start} : \text{vtokens end}] )) \\} \\}
> $$
>
> Here, each block $ B $ is processed independently, and the results are combined using incremental normalization.
>
> [1] Dao, T. (2022). Flashattention: Fast and memory-efficient exact attention with io-awareness. Advances in Neural Information Processing Systems.
>
> [2] Dao, T. (2023). Flashattention-2: Faster attention with better parallelism and work partitioning. arXiv preprint arXiv:2307.08691.

---

> > ### Author Response · Authors · 2024-11-28
> >
> > Dear reviewer hiZv, we have updated the solution to the compatibility issue between Flash Attention and our method that you were particularly concerned about in the appendix. We are looking forward to your feedback!

---

> ### Author Response · Authors · 2024-11-23
> **Response to Reviewer hiZv (Part 2 / 3)**
>
> ---
>
> > 2. More explanations and discussions about the performance of SparseVLM.
>
> Thank you for your insightful comments. Firstly, we want to clarify that there is an efficiency-performance trade-off in token sparsification, and under the same efficiency, our method already obtains state-of-the-art performance compared to other methods. Moreover, the "retain 192 tokens" is too aggressive to have a minor impact on performance; on the other hand, one can obtain marginal performance drops by setting moderate sparsification ratios. As demonstrated by the additional experiments in the table below, retaining mid-range 448, 384, and 320 tokens can preserve the performance while still having obvious efficiency gains. For instance, with 22% of the vision tokens deleted (448 tokens in the table), the average performance drop is only 0.5%.
>
> Note that this trade-off is adaptively controllable by our proposed sparsification level adaptation, which means that one can easily set the sparsification ratio to achieve the desired efficiency and performance.
>
> | Tokens | MME |  POPE | TextVQA
> | :--------  | :-----  | :----:  | :----:  |
> | 576 (original) | 1862| 85.9 | 58.2
> | 448 | 1845 (**99.1%**) |85.9 (**100%**) | 57.8 (**99.3%**)
> |384 | 1796 (**96.5%**) | 85.8 (**99.9%**) | 57.7 (**99.1%**)
> | 320 | 1778 (**95.5%**) | 85.2 (**99.2%**) | 57.6 (**99.0%**)
>
> ---
>
> > 3. The comparison of our SparseVLM with baseline matched with FlashAttn.
>
> Thank you for your detailed suggestions. Here, we utilized the mentioned SparseVLM FlashAttn in Part 1 compared with the baseline matched with FlashAttn. We conducted comprehensive experiments on LLaVA and MGM across three benchmarks: POPE, TextVQA, and MME. The results are shown in the following table. Besides, we compared our method with the random sparse matched with FlashAttn, and we observed that our method has significant improvement, when under a similar CUDA time.
>
>
> | Method                          | POPE (Avg Acc) | Avg CUDA Times (ms) | TextVQA (Avg Acc) | Avg CUDA Times (ms) | MME (Avg Acc) | Avg CUDA Times (ms)| Avg TFLOPs |
> |---------------------------------|----------|-----------------|-------------|--------------------|---------|----------------|------------|
> | Original LLaVA w/ Flash (576)   | 85.88    | 427696.68     | 58.21       | 286758.96       | 1862    | 125205.35    | 4.37       |
> | LLaVA (random Sparse w/ Flash)  | 84.67    | 314391.58     | 55.64       | 215478.40         | 1803    | 94158.56    | 2.25       |
> | **LLaVA (sparseVLM w/ Flash)**      | 85.21    | 315236.49     | 57.51       | 212753.22        | 1835    | 96313.14    | 2.24       |
> | Original MGM w/ Flash (576)     | 85.73    | 441471.83     | 64.98       | 294506.99        | 1842    | 129139.07    | 4.37       |
> | MGM (random Sparse w/ Flash)    | 83.32    | 351456.66     | 61.65       | 213259.19        | 1820    | 88876.37    | 2.40       |
> | **MGM (sparseVLM w/ Flash)**        | 84.57    | 351399.50     | 63.95       | 211810.73        | 1845    | 88883.89    | 2.39       |
>
> ---
>
> > 4. The report of Qwen2-VL with SparseVLM on high-resolution image understanding benchmarks.
>
> Thank you for your valuable suggestion. We applied our SparseVLM to Qwen2-VL and conducted comprehensive experiments on InfoVQA, and AI2D. The results are shown in the following table. We found that our method still shows superiority on high-resolution image understanding benchmarks. For instance, on the AI2D, when the sparsification ratio increases from 43.74% (768 tokens) to 81.25% (256 tokens), SparseLLaVA only drops the accuracy by 5% without any additional training, while reducing the CUDA Times and TFLOPs by 42.75% and 75.09% respectively. This demonstrates that high-resolution image understanding benchmarks like AI2D also contain a significant amount of token redundancy.
> Due to the limitations of computing power and time, we did not conduct experiments on DocVQA for the time being (takes two to three hours), and this will be accomplished in our revised version.
>
> ### AI2D_TEST Performance Comparison
>
> | Tokens           | Accuracy | CUDA Times (ms)            | TFLOPs           |
> |------------------|----------|-------------------------|------------------|
> | 1365 (w/o sparse)| 79.27    | 350464.81      | 10.64 |
> | 768              | 77.23    | 290784.72       | 6.234  |
> | 512              | 76.13    | 244586.19       | 4.49 |
> | 256              | 73.25    | 200624.51      | 2.65 |
>
> ### InfoVQA_Test Performance Comparison
>
> | Tokens              | Accuracy | CUDA Time (ms)        | TFLOPs           |
> |---------------------|----------|-----------------------|-------------------|
> | 3979 (w/o sparse)   | 76.43    | 1040070.35    | 34.69 |
> | 2485      | 70.94    | 821925.11     | 20.34 |
> | 2213     | 70.09  | 770971.13     | 18.18 |
> | 1778     | 67.34   | 694791.19     | 14.82 |

---

> ### Author Response · Authors · 2024-11-23
> **Response to Reviewer hiZv (Part 3 / 3)**
>
> ---
>
> > 5. The explanation of some hyperparameters.
>
> Thank you for your valuable comments, and we explain them in detail.
>
> (1) The m in equation (6) is the mean of the $R$ matrix, a dynamical number determined by $R$ instead of a specific and constant one.
>
> (2) $\lambda$ in equation (8) is a scaling factor to determine the number of deletions, and it is up to your needs. For instance, if you want to sparsify the number of vision tokens to 192 on MME, the $\lambda$ should be set to 13.5; if you want to sparsify the number of vision tokens to 64 on TextVQA, the $\lambda$ should be set to 0.8.
>
> (3) The $\tau$ in equation (9) is a recycling ratio to decide the number of reconstruction candidates, which is a constant number. In all our experiments, we set it to 30%. Although it is not well-designed, our method is still effective and reserves more information with fewer slots.
>
> ---
>
> > 6. How does SparseLVM insert these recycled tokens into the preserved tokens?
>
> Firstly, whenever irrelevant tokens are removed or new tokens are inserted, our method reconstructs the positional encodings to ensure that the spatial relationships remain unchanged. Specifically, the recycled tokens are added at the end to preserve the original positional relationships. Furthermore, previous methods like [1][2] also adopt a similar core idea to reconstruct positional encodings for the insertion of new tokens, while still keeping the performance of their methods. Finally, these recycled tokens can be seen as another form of system or [cls] tokens, which contain the semantic information of multiple tokens.
>
> [1] Shang, Yuzhang, et al. "Llava-prumerge: Adaptive token reduction for efficient large multimodal models." arXiv preprint arXiv:2403.15388 (2024).
>
> [2] Zeng, Wang, et al. "Not all tokens are equal: Human-centric visual analysis via token clustering transformer." Proceedings of the IEEE/CVF Conference on Computer Vision and Pattern Recognition. 2022.

---

> ### Author Response · Authors · 2024-11-25
> **Discussion to Reviewer hiZv**
>
> We sincerely thank you for your efforts in reviewing our paper. We have provided corresponding responses and results, which we believe have covered your concerns. We hope to further discuss with you whether your concerns have been addresses or not. Please let us know if you still have any unclear part of our work.
>
> Best, Authors

---

> ### Author Response · Authors · 2024-12-02
> **Gentle Reminder Regarding Review of Reviewer hiZv**
>
> **Dear reviewer hiZv**,
>
> I hope this message finds you well. We greatly appreciate the valuable feedback and suggestions you have provided so far. As the deadline approaches, we are eager to receive your feedback on our response and revisions. If possible, we kindly request an update on the progress of the review to ensure we can address any further comments or revisions promptly.
>
> Should you require any additional information or assistance from our end to help facilitate the review process, please do not hesitate to let us know. Your insights are highly valuable to us, and we genuinely appreciate your time and effort in reviewing our paper.
>
> Thank you for your patience and cooperation. We are looking forward to hearing from you soon.
>
> Warm regards,
>
> Submission402 Authors.

---

### Official Review · Reviewer_RuY4 · 2024-11-06

**Soundness:** 2
**Presentation:** 3
**Contribution:** 3
**Rating:** 6
**Confidence:** 4

**Summary:**

The paper introduces SparseVLM, a method to accelerate vision-language models (VLMs) by prunning vision tokens incrementally over layers, based on its significance for (a subset) the text tokens. The set of (significant) visual tokens to keep is computed from the self-attention scores in each layer, and the set of relevant text tokens is computed just once, using the dot-product between the text and image tokens after being embedded to the same size. This tries to reduce the computational overhead of the method, achieving real wallclock time speed-ups, for different prunning levels. The authors also propose to aggregate and reconstruct some tokens, to prevent completely losing the information of the tokens that are decided to prune.
The paper presents results in different image and video understanding benchmarks, and compares the proposed method against two recent baselines (ToME and FastV). The results show that the proposed method improves over these baselines across different prunning thresholds, and achieves significant memory, FLOP and runtime reduction with roughly the same accuracy, when compared to FastV.

**Strengths:**

- The proposed method is compared against two recent and popular baselines: ToMe and FastV.
- The paper is well structured and the method is well explained (modulo some typos / ambiguous equations, see weaknesses).
- The method does not require any type of fine-tuning, so it can be used on top of different VLMs, which broadens its potential adoption.
- For the same number of retained tokens, Table 1 and Table 2 show that the proposed method represents a huge accuracy improvement respect the baselines.
- The paper ablates the use of token reconstruction in Table 3, which shows that the proposed improvement significantly improves over the core SparseVLM method.

**Weaknesses:**

- The results in Table 1 and Table 2 do not reflect the reduction in neither FLOP, nor wallclock runtime. Only Table 4 offers some results comparing the proposed method only against FastV. However it's not clear to which benchmark(s) the reported accuracy corresponds to. Also, the baseline storage memory is missing (although it can be calculated from the remaining values). I would suggest that the authors report a similar figure to Figure 4 with two plots showing the avg accuracy w.r.t. a) letency or total runtime and b) FLOPs. This would represent much better the cost-vs-quality tradeoffs, for SparseVLM applied on both LLaVA and MGM. This is crucial to increase the rating of the paper. If space is limited, I would suggest reporting values for individual datasets in the appendix, and report only the average in the main text (unless there are any major outliers).
- Table 2 does not even represent the speed-vs-accuracy trade-off, nor the "token budget"-vs-accuracy, since only a single token budget of 135 is represented. Also, this value is not the same used in any of the image results reported in Table 1. Which begs the question: why was this particular value used? Please, provide figures as described above.
- It's not 100% clear how $\mathbf{P}$ in section 3.2 is calculated. According to eq. (1) and (2), $\mathbf{P}$ is a subset of rows and columns of the attention matrix (after the softmax), but lines 183-184 refer the "logits" (i.e. $\frac{\mathbf{Q}\mathbf{K}^\top}{\sqrt{D}}$). It's also not clear if the attention matrix is re-normalized after the selected visual tokens are dropped from the keys or not.
- Notation in eq. (7) is ambiguous. The index $j$ in the sum isn't used anywhere. Also, notice that the size of $\mathbf{H}_v \mathbf{H}_q^\top$ is $L_v \times L_q$, which is inconsistent with the sum over $j$, assuming $j$ denotes a column index, since $\mathbf{R}_i$ supposed to be the average over visual tokens for the $i$-th query token. This is a small mistake that can be fixed by using $\mathbf{H}_q \mathbf{H}_v^\top$, to match the dimension order of $\mathbf{P}$ is $L_v \times L_q$ (i.e. text $\times$ vision).
- The choice of the relevant text token select threshold $m = \text{Mean}(\mathbf{R})$ isn't justified. Why this threshold and not something else? E.g. the text tokens in the with the highest $R$ score such that the sum of
- The number of vision tokens to prune is based on the rank of $\textbf{P}$, this can be problematic due to numerical precision. For instance, suppose that $n = L_t = L_v$, what happens if we get that half of the the singular values of P are $10^{-5}$ and the rest are $10^5$? The rank would be technically $n$, but is it really or do we get $10^{-5}$ rather than 0 due to numerical errors?

**Questions:**

- Suggestion: Instead of using the rank, an alternative approach (which is not prone to the numerical issues that I discussed above), is to prune based on the (relative) singular values of $\mathbf{P}$:
    1) First compute the singular values of $\mathbf{P}$, assume that these are returned in decreasing order.
    2) Divide each value by the total sum of singular values (a.k.a. the "energy" of the matrix). Let's call this vector $E$, i.e. the relative energy of each singular value.
    3) Prune $N - k$ tokens, where $k$ is the smallest value such that $\sum_{i=1}^k E_i \geq \lambda$.
- Could you please add the memory used by the baseline in Table 4?

---

> ### Author Response · Authors · 2024-11-23
> **Response to Reviewer RuY4 (Part 1 / 2)**
>
> We sincerely thank the reviewer RuY4 for the efforts in reviewing our paper. Our responses according to the reviewer's comments are summarized as follows.
>
> ---
>
> > 1. The additional efficiency experiments and explanation about latency and FLOPs on image understanding tasks.
>
> Thanks for the suggestion. We have complemented more experiments and revised the paper accordingly.
>
> (1) **Baseline Storage Memory**: The revised version of the manuscript updates the baseline value to 302.4.
>
> (2) **More Trade-off Figures**: Thank you for your detailed suggestions. We have included latency-vs.-accuracy and FLOPs-vs.-Accuracy tradeoffs for SparseVLM applied to LLaVA and MGM across three benchmarks: POPE, TextVQA, and MME. Figures 9 and 10 in Appendix A.7 now include a total of 12 sub-figures illustrating these tradeoffs on the image understanding tasks.
>
> (3) **Efficiency Summary**: The following table presents the latency-vs.-average accuracy and FLOPs-vs.-Average accuracy results for individual datasets, providing a comprehensive overview of our method's efficiency.
>
> ### latency-vs.-average accuracy
>
> | Model                | POPE (Acc)    | CUDA Times (ms)   | TextVQA (Acc)    | CUDA Times (ms)   | MME (Acc)    | CUDA Times (ms)   |
> |----------------------|--------|-------------|--------|-------------|--------|-------------|
> | LLaVA(sparseVLM)     | 85.2  | 315236.49 | 57.5  | 212753.22 | 1834.5 | 96313.14 |
> | LLaVA(random Sparse) | 84.6  | 314391.58 | 55.6  | 215478.40  | 1803.0   | 94158.56 |
> | MGM(sparseVLM)       | 84.7 | 351399.50 | 64.0  | 211810.73 | 1845.5| 88883.89 |
> | MGM(random Sparse)   | 83.2 | 351456.66 | 61.5  | 213259.19 | 1819.5| 88876.37 |
>
> ### TFLOPs-vs.-average accuracy
>
> | Model                | POPE (Acc)    | TFLOPs      | TextVQA (Acc)    | TFLOPs      | MME (Acc)    | TFLOPs      |
> |----------------------|--------|-------------|--------|-------------|--------|-------------|
> | LLaVA(sparseVLM)     | 85.8 | 2.08 | 57.4 | 2.53 | 1797.0   | 2.13 |
> | LLaVA(random Sparse) | 83.9 | 2.11 | 53.6  | 2.54 | 1747.6 | 2.12 |
> | MGM(sparseVLM)       | 84.7  | 2.46 | 63.5  | 2.56 | 1837.8 | 2.15 |
> | MGM(random Sparse)   | 82.3 | 2.47  | 58.6  | 2.58  | 1798.8 | 2.16 |
>
> In summary, the above experiments fully demonstrate the effectiveness of our method in reducing latency and computational complexity.
>
> ---
>
> > 2. The additional efficiency experiments and explanation about latency and FLOPs on video understanding tasks.
>
> Thank you for your valuable feedback regarding video understanding tasks.
>
> (1) **Clarification of Token Budget**: The 135 is the equivalent number of tokens after our pruning and recycling. Specifically, the original 2056 tokens are pruned in the second layer to a single token. For the equivalent calculation: the first two layers without pruning, have 2 × 2056 tokens, while the subsequent 30 layers, after pruning, contribute 30 × 1 token. With our SparseVLM performing token recycling, the average number of tokens increases slightly to 135. To ensure a fair comparison, we align the token count in FastV with this average.
>
> (2) **More Trade-off Figures**: In our revised manuscript, we have included additional figure (Figure 11 in Appendix A.7) that demonstrate both the speed-versus-accuracy and token budget-versus-accuracy trade-offs across multiple token budgets on TGIF and MSVD datasets, as summarized in the table below. This will provide a clearer understanding of our method's performance and its implications.
>
> ### Video-LLaVA Trade-offs
>
> | Method | TGIF  (Acc)          | CUDA Times (ms) | MSVD (Acc) | CUDA Times (ms) | TFLOPs (Avg.) |
> | :--------------------  | :--------------- |:--------  | :-----  | :----- | :----- |
> | **135 tokens**          |           |               |          |               |          |
> | FastV| 23.1 2.47| 920786.17 | 38.0 2.71 | 532909.58 | 1.21 |
> | SparseVLM | 44.7 3.29| 1878079.03 | 68.2 3.90 | 924956.49 | 1.21 |
> | **700 tokens**          |           |               |          |               |          |
> | FastV| 21.2 2.53| 3053116.46 | 68.6 3.87 | 1581100.35 | 5.03 |
> | SparseVLM | 46.3 3.32 | 3763170.63 | 69.9 3.93 | 1966915.04 | 5.03 |
> | **1092 tokens**          |           |               |          |               |          |
> | FastV| 25.5 2.74| 4418518.86 | 70.0 3.9 | 2294790.37 | 7.76 |
> | SparseVLM | 46.4 3.33 | 5514788.47 | 69.9 3.93| 2738813.35 | 7.76 |

---

> > ### Comment · Reviewer_RuY4 · 2024-11-26
> >
> > Thank you so much for the additional clarifications regarding runtime and FLOPs.
> >
> > If I read the last table correctly, SparseVLM does achieve better results than FastV, but it's at the expense of a near 2x slower model (but same FLOPs). This is concerning, since one could argue that training FastV for 2x the number of steps, or tweaking some other hyperparam that makes the model essentially 2x slower, could achieve better results than the proposed method.
> >
> > An argument in favor of the proposed method would be some potential optimization that was done in FastV but not SparseVLM. Can you provide more context?

---

> ### Author Response · Authors · 2024-11-23
> **Response to Reviewer RuY4 (Part 2 / 2)**
>
> ---
>
> > 3. The explanation for the calculation of $P$.
>
> We appreciate your valuable comments regarding the definition of $P$. (1) $P$ is a subset of the attention matrix selected by vision and text indexes, derived **after applying softmax**. The reference to 'logits' in line 183 is indeed inappropriate, as logits refer to unnormalized values. We have fixed it in the revised version. (2) Once $P$ is obtained, we do not perform normalization again, as it has already undergone softmax.
>
> ---
>
> > 4. The explanation for the eq. (7).
>
> Thank you for bringing this to our attention and pointing out the error in equation (7). We have revised the notation as follows:
>
> $$    R = \\frac{1}{L_v} \\sum_{j=1}^{L_v} (\text{Softmax} (H_v {H_q}^T))[j,:], $$
>
> Where is $H_vH_q^{T} \in \mathbb{R}^{L_v \times L_q}$. We first take the mean over the vision dimension, leaving only the text dimension, where it is $R \in \mathbb{R}^{L_q}$. Then, we take the mean over the text dimension to generate the threshold of raters $m$.
>
> ---
>
> > 5. The explanation for text token select threshold.
>
> Thank you for your insightful comments and advice. In our method, we do not adopt the complex design for the threshold, and only utilize an average tool to generate it, which is simple but effective. Furthermore, based on your suggestion, we conducted additional ablation experiments on LLaVA to compare the accuracy differences between using the mean and the topK ($k$=8) on various benchmarks, and the results are as follows:
>
> | Method  | MME (192 tokens) | POPE (128 tokens) | TextVQA (64 tokens)
> | :--------  | :-----  |:--------  | :-----  |
> | topk | **1731**| 79.9 | 51.6
> |mean|  1721 | **80.5** |**51.8**
>
> Although topK selection has a minimal improvement on MME, we argue that adaptive selection by the mean is general and robust to various prompts compared with specific topK selection.
>
> ---
>
> >6. The discussion of the number of vision tokens to prune is based on the rank of $P$.
>
> Thank you for the insightful question and suggestion!
>
> (1) **Clarification of extreme cases**: The situation mentioned above hardly occurs during the actual inference process, because the values in the attention matrix after applying softmax will reduce the difference between larger and smaller values, resulting in a smoother distribution of the matrix.
>
> (2) **Reproduction of your suggestion**: First of all, thank you for your valuable suggestion, and we applied it to SparseVLM for experiments on TextVQA, MME, and POPE. As shown in the table below, when the number of singular values with higher contributions identified after Singular Value Decomposition (SVD) matches the number of vision tokens retained after the Rank(P) operation, the accuracy of both methods remains largely similar. However, we note that the CUDA execution time for our method is slightly faster than that of SVD.
>
> (3) **Optimization for extreme cases**: To summarize, the differences in accuracy between the two methods are minimal, suggesting that the numerical accuracy issue you mentioned rarely arises during actual reasoning. Besides, to completely avoid the situation you mentioned, we will set values smaller than $10^{-5}$ to zero in advance. Subsequently, the number of non-zero singular values will be used as the rank for selecting vision tokens.
>
> | Method                  | MME       | CUDA Times (ms) | POPE      | CUDA Times (ms) | TextVQA   | CUDA Times (ms) | Avg TFLOPs|
> |-------------------------|-----------|-----------|-----------|-----------|-----------|---------------|-----------|
> | **192 tokens**          |           |               |          |               |          |               |          |
> | SparseVLM with Rank     | **1735.1**| **108215.18** |**83.36** | **395540.04** |**56.63** | **260698.30** | **1.88** |
> | SparseVLM with SVD      | 1723.8    | 129365.52     |82.77     | 399198.76     |56.43     |  261643.33    | **1.88** |
> | **128 tokens**          |           |               |          |               |          |               |          |
> | SparseVLM with Rank     | **1712.9**| **100933.74** |**80.12** | 374270.92     |55.37     | 243447.59     | **1.47** |
> | SparseVLM with SVD      | 1697.5    | 109545.98     |80.05     | **373136.85** |**55.40** | **242088.48** | **1.47** |
> | **64 tokens**           |           |               |          |               |          |               |          |
> | SparseVLM with Rank     | **1507.1**| **91709.23**  |**75.07** | **342154.00** |**51.53** | **210759.78** | **1.06** |
> | SparseVLM with SVD      | 1469.6    | 95119.82      |74.87     | 348250.22     |51.48     | 222213.86     | **1.06** |
>
> ---
>
> > 7. Add the memory used by the baseline in Table 4.
>
> Thank you for your valuable suggestion regarding the memory usage for the baseline in Table 4, whose value is 302.4. We have updated it in the revised version.

---

> > ### Comment · Reviewer_RuY4 · 2024-11-26
> >
> > I really appreciate all the effort that the authors put in ablating my suggestions and updating the manuscript fixing typos.
> >
> > I believe that all of these questions/comments/concerts (questions 3-7) have been adequately addressed, and I'll be happy to increase my score accordingly (but notice that there I still have some concerns regarding question 2).

---

> ### Author Response · Authors · 2024-11-25
> **Discussion to Reviewer RuY4**
>
> Dear Reviewer RuY4,
>
> We sincerely thank you for your efforts in reviewing our paper. We have provided corresponding responses and results, which we believe have covered your concerns. We hope to further discuss with you whether your concerns have been addresses or not. Please let us know if you still have any unclear part of our work.
>
> Best,
> Authors

---

> ### Author Response · Authors · 2024-11-28
> **Further discussion regarding question 2**
>
> We greatly value your feedback!
>
> To sparstify the numerous vision tokens in video understanding tasks, we inquire about the necessity of sparse operations **for each layer**. This leads to **multiple rank operations being executed**, consequently increasing the time consumption. The manner differs from the strategy FastV employs, where sparse operations are only **conducted in a single layer** (Layer 1). To address this, aligned with the FastV, we have optimized the sparsification strategy in video understanding tasks and conducted sparsification in only three layers (Layers 1, 4, 15), and performed swift validation experiments by randomly sampling subsets from TGIF and MSVD datasets. The results are listed as follows and visualization has been updated in the Appenidx. We observed that our method has significant improvement over FastV under comparable CUDA time constraints. For instance, our approach achieves even **higher accuracy** than FastV in the 1092-token setting and also demonstrates **a significant speed advantage**, especially in the MSVD dataset using the 700-token setting.
>
> | Method | TGIF [1000 Cases]   | CUDA Times (ms) | MSVD [1000 Cases] | CUDA Times (ms) | Avg TFLOPs |
> | :--------------------  | :--------------- |:--------  | :-----  | :----- | :----- |
> |                             | Acc /  Score |  | Acc / Score |          |         |
> | **135 tokens**          |           |               |          |               |          |
> | FastV [Layers 2]| 9.3 / 2.10 | 48720.41 | 44.1 / 3.00 | 48764.48 | 1.21 |
> | SparseVLM  [Layers 2,5,16]| 12.81 / 2.27| 50791.69 | 53.9 / 3.41 | 50543.91 | 1.21 |
> | **700 tokens**          |           |               |          |               |          |
> | FastV [Layers 2]| 17.4 / 2.45 | 104383.42 | 68.8 / 3.88 | 105897.09 | 5.03 |
> | SparseVLM [Layers 2,5,16]| 18.7 / 2.50 | 110643.87 | 71.2 /  3.95 | 111168.20 | 5.03 |
> | **1092 tokens**          |           |               |          |               |          |
> | FastV [Layers 2]| 17.9 / 2.48| 164239.53 | 69.6 / 3.87 | 165173.51 | 7.76 |
> | SparseVLM [Layers 2,5,16]| 18.9 / 2.53 | 167492.27 | 71.7 / 3.95| 166668.26 | 7.76 |

---

> > ### Author Response · Authors · 2024-12-02
> > **Polite Reminder Regarding Score Update for Our Response**
> >
> > **Dear Reviewer RuY4**,
> >
> > I hope this message finds you well. I wanted to express my gratitude for your positive feedback on our response and for considering an increase in the rating for our submission.
> >
> > If it is convenient for you, could you kindly update the rating based on your most recent assessment? Your feedback and evaluation are crucial to us, and we appreciate your time and effort in reviewing our work.
> >
> > Thank you once again for your consideration. We look forward to any updates you might provide.
> >
> > Warm regards,
> >
> > Submission402 Authors

---

### Author Response · Authors · 2024-11-23
**General Response**

Dear reviewer RuY4, hiZv, YHTW, 7sbX, and Rbrk

We sincerely thank you for your valuable and constructive comments. We have provided detailed answers and revised the paper accordingly. Specifically, it includes the following aspects:

1. **Expanded Efficiency Experiments**: We have incorporated comprehensive experiments such as latency-vs.-accuracy and FLOPs-vs.-Accuracy analyses for image (e.g., LLaVA and MGM) and video (e.g., VideoLLaVA) understanding architectures across diverse benchmarks, with a detailed examination of memory costs.

2. **In-depth Specific Tasks Experiments**: We have augmented our study with additional experiments focusing on logical reasoning and higher-resolution tasks. Furthermore, we have implemented SparseVLM in the Qwen-VL architecture to validate its generality.

3. **Detailed Performance Discussion**: Our methodology shows state-of-the-art performance at efficiency levels comparable to other existing methods. The adaptability of our approach is highlighted through the proposed sparsification level adaptation, allowing users to easily configure the sparsification ratio for desired efficiency-performance trade-offs.

4. **Supplementary Ablation Experiments**: We have included further comparisons between rank and SVD, explored additional token settings for testing text raters, and conducted more experiments regarding text token selection thresholds.

5. **Enhanced Explanations**: We have provided elaborations on Equations (6), (7), (8), and (9), elucidated the strategy for position embedding, and demonstrated our advantages in multi-turn conversations.

As most reviewers were concerned about **the compatibility of SparseVLM with FlashAttention**, we have provided a general response: Our exploitation of the attention matrix is to compute the token sparsification, which is similar to the self-attention operation. Therefore, we can readily utilize FlashAttention to accomplish this by employing a special values matrix in self-attention. We have included the detailed implementation in the individual responses. As we only perform sparsification in a few layers in the LLM, the additional use of FlashAttention will not bring significant computation overhead. We have provided a more detailed experimental analysis in the revised version.

Additionally, we have revised the paper and highlighted the changes in blue fonts. Below is a summary of the updates:

1. Line 67-68, update the caption in Figure 1(c).
2. Line 123-124, update the number of image tokens.
3. Line 143-145, update the description of the ToMe method.
4. Line 215, update the equation (7).
5. Line 450, add the memory used by the baseline in Table 4.
6. In A.7, add more trade-off results in Figures 9, 10, and 11.

Thank you once again for your valuable feedback and suggestions, which have significantly improved the quality of our work. If you have any further questions or if anything remains unclear, please don’t hesitate to let us know. We would be more than happy to discuss your concerns in greater detail. If the response is satisfactory, we hope you will raise the score so that more people can see this paper.

---

### Meta-Review · Area_Chair_cTF5 · 2024-12-21

**Metareview:**

The authors address a timely and relevant problem: The number of visual tokens to a VLM are often redundant and substantially increase the computational cost of the model. The authors propose to address this by proposing an iterative token sparification strategy that reduces the overall number of tokens in a VLM, by conditioning on the input question, inspired by previous works in the literature on token reduction such as ToMe and FastV. Moreover, the approach does not require any training, which means that it can be potentially applied to a wide range of VLMs.

Reviewers had multiple concerns in the rebuttal, some of which the authors addressed convincingly: For example, the proposed approach is still comparible with FlashAttention.

However, a remaining concern is that the proposed sparsisfication strategy still requires computation itself. Therefore, comparing to prior works based only on the number of tokens as in Table 1, is not sufficient. The authors should transparently compare the runtime and GFLOPs as well, as raised by some reviewers (It has also been widely reported [before](https://arxiv.org/abs/2110.12894) that reporting only specific efficiency indicators can be misleading). Although the authors provided these values in the rebuttal, the results are underwhelming for a range of token budgets. Overall this is concerning, as well as the fact that this, and subsequent replies to Reviewer-RuY4 are not included in the revised main paper.

Reviewer-YHTW also mentioned three more recent papers which achieve better results than FastV which are not compared to. The AC and Senior-AC agree that Victor[1] and UPop[2] are not fair comparisons. But for PDrop, the authors have once again not compared the actual runtime and GFLOPs.

Therefore, after some deliberation, the AC and Senior-AC decided to reject the paper. Authors are encouraged to revise their paper according to the reviewers' feedback, in particular comparing fairly to prior works not only in the token budget, but the FLOPs and runtime as well, and to resubmit to another venue.

**Additional Comments On Reviewer Discussion:**

Refer to the above. The rebuttal addressed some of the reviewers concerns (ie that the proposed method is compatible with flash attention). However, other concerns were not well addressed in the rebuttal: When comparing to prior works based in terms of runtime and GFLOPs, the results are underwhelming for a range of token budgets.

---

### Decision · Program_Chairs · 2025-01-22

Reject